# A draft genome of Drung cattle reveals clues to its chromosomal fusion and environmental adaptation

Yan Chen[1,7], Tianliu Zhang[1,7], Ming Xian[1], Rui Zhang[1], Weifei Yang[2,3], Baqi Su[4], Guoqiang Yang[5], Limin Sun[6], Wenkun Xu[6], Shangzhong Xu[1], Huijiang Gao[1], Lingyang Xu[1], Xue Gao [1✉] & Junya Li [1✉]

Drung cattle (*Bos frontalis*) have 58 chromosomes, differing from the *Bos taurus* $2n = 60$ karyotype. To date, its origin and evolution history have not been proven conclusively, and the mechanisms of chromosome fusion and environmental adaptation have not been clearly elucidated. Here, we assembled a high integrity and good contiguity genome of Drung cattle with 13.7-fold contig N50 and 4.1-fold scaffold N50 improvements over the recently published Indian mithun assembly, respectively. Speciation time estimation and phylogenetic analysis showed that Drung cattle diverged from *Bos taurus* into an independent evolutionary clade. Sequence evidence of centromere regions provides clues to the breakpoints in BTA2 and BTA28 centromere satellites. We furthermore integrated a circulation and contraction-related biological process involving 43 evolutionary genes that participated in pathways associated with the evolution of the cardiovascular system. These findings may have important implications for understanding the molecular mechanisms of chromosome fusion, alpine valleys adaptability and cardiovascular function.

[1] Laboratory of Molecular Biology and Bovine Breeding, Institute of Animal Science, Chinese Academy of Agricultural Sciences, 100193 Beijing, P.R. China. [2] 1 Gene Co., Ltd, 310051 Hangzhou, P.R. China. [3] Annoroad Gene Technology (Beijing) Co., Ltd, 100176 Beijing, P.R. China. [4] Drung Cattle Conservation Farm in Jiudang Wood, Drung and Nu Minority Autonomous County, Gongshan, 673500 Kunming, Yunnan, P.R. China. [5] Livestock and Poultry Breed Improvement Center, Nujiang Lisu Minority Autonomous Prefecture, 673199 Kunming, Yunnan, P.R. China. [6] Yunnan Animal Husbandry Service, 650224 Kunming, Yunnan, P.R. China. [7] These authors contributed equally: Yan Chen, Tianliu Zhang. ✉email: gaoxue@caas.cn; lijunya@caas.cn

Drung cattle is named for its unique distribution in the Drung and Nujiang river basins of Yunnan Province in China[1]. This unique semi-wild endangered bovine species is commonly characterized by white head and white stockings on four white legs (Supplementary Fig. 1a–e), exhibiting a similar appearance to Thai gaur, Indian gaur and Malayan gaur. In the biological taxonomy, Drung cattle belong to the species *Bos frontalis*, genus *Bos*, group *Bovina*, tribe *Bovini*, subfamily *Bovinae*, the family *Bovidae*[2], also known as gayal or mithun. However, the origin and evolution history of Drung cattle has hitherto not been proven conclusively. The distinctive phenotypic traits and specific karyotype ($2n = 58$) distinguish gayal from gaur (*Bos gaurus*; $2n = 56$) and cattle (*Bos taurus* and *Bos indicus*; $2n = 60$)[3,4]. The common view is that a Robertsonian translocation, Rob (2;28), led to a *B. frontalis* diploid number of the $2n = 58$ karyotype[4,5]. On the other hand, Drung cattle inhabit the typical alpine valley and subtropical rain forests at an altitude of 400–4000 m, which gives them unique physiological features that equip them for life in complex terrain and humid environments. Studies have shown that the hemoglobin levels in Drung cattle, as well as the number of red and white blood cells, were equivalent to those of yaks living on the Tibetan Plateau[6,7]. Compared with the other native cattle in Yunnan, the muscle fibers of Drung cattle are smaller in diameter, higher in density, and larger in number (Supplementary Fig. 2a, b). Moreover, a significant difference was found to be the intervening thicker connective tissues in the myocardium of Drung cattle surrounded by abundant blood vessels and numerous capillaries (Supplementary Fig. 2c, d). However, the genetic basis of its chromosome evolution and adaptability remains largely unknown.

To date, the genomes of several bovine species under the tribe *Bovini*, including *B. taurus* (cattle)[8], *B. indicus* (zebu)[9], *Bison bison* (American bison)[10], *Bison bonasus* (wisent)[11], *Bos grunniens* (yak)[12], *Bubalus bubalis* (water buffalo)[13–15] and *Syncerus caffer* (African buffalo)[16,17], have been sequenced and annotated (Supplementary Data 1). High-quality reference genome sequences and gene annotations can be used as genomic resources to facilitate further functional and evolutionary studies. Recently, the draft genome of *B. frontalis* was released from a female Drung cattle using the Illumina 2000 platform[18] and a female India mithun using Illumina HiSeq and PacBio technology[19]. In comparison with the abovementioned bovine species, the two published genome assemblies of *B. frontalis* had relatively lower completeness, especially the smaller sizes of scaffold N50 and contig N50 genome sizes[15]. It is well known that the main factors affecting the accuracy and continuity of the assembly include the length of reads, size of the library, the accuracy of reads, uneven sequence depth, and complexity of the genome[20,21]. Advances in sequence technology, assembly approaches, quality control, and bioinformatics strategies are continually being developed to overcome these problems[22]. The emergence of innovative, disruptive technologies such as long-read sequence, linked reads, Hi-C, and optical mapping has greatly driven the improvement of a high-quality genome assembly[23–25]. Currently, there are many advantages to using hybrid de novo assembly approaches than those produced by short-read sequence alone to improve the quality of genome assemblies.

Herein, we performed a genome sequence of a male Drung cattle by adopting a hybrid de novo assembly approach using a combination of short reads from Illumina HiSeq & MiSeq platform, long reads from PacBio platform, and linked-reads from 10× Genomics platforms to generate an improved *B. frontalis* genome assembly. We then analyzed centromeric satellite repeats to discern potential fusion regions of Robertsonian translocation in *B. frontalis*. Finally, a combined analysis of genomics, transcriptomic, and DNA methylation of Drung cattle was carried out to identify functional genes and to understand the molecular biological mechanisms associated with animal adaptation in the challenging ecology of alpine valleys.

## Results

**Sequencing and assembly of the Drung cattle genome**. To perform de novo assembly of the Drung cattle genome, three different technologies, short-read Illumina sequencing, long-read PacBio sequencing, and linked-read 10× Genomics sequencing, were combined (Supplementary Fig. 3). First, we generated ~235-fold coverage of Illumina paired-end reads (633.7 Gb) at three different short insert sizes (200 bp, 450 bp, 800 bp), ~133-fold coverage of Illumina mate-pair reads (203.5 Gb) at four long insert sizes (2 kb, 5 kb, 10 kb, 20 kb) and ~11-fold coverage of PacBio sequencing data (30.9 Gb). Next, to order and orient the scaffolds into longer blocks, ~75-fold barcoded sequence coverage was obtained using the 10x Genomics Chromium platform (Supplementary Data 2). The total Drung cattle genome size of the final assembled version (Drung_*v*1.2) was 2.74 Gb (Supplementary Data 3), which approximated the estimated genome size of 2.72 Gb by flow cytometry (Supplementary Fig. 4 and Supplementary Data 4) and 2.67 Gb by *k*-mer analysis (Supplementary Fig. 5).

To evaluate the completeness and integrity of our assembly, we found that 98.87% of the raw paired-end reads generated from the small insertion libraries were aligned to our final assembled genome. In addition, BUSCO (*v*2.0, Benchmarking Universal Single-Copy Orthologs) analysis of the draft assembly was run against the 4,104 mammalian ortholog database. The genome was 92.05% complete, with 91.08% single, 0.97% duplicated, 3.85% fragmented, and 4.09% missing copies. These scores were comparable to those of other bovine genomes (Supplementary Data 5). Furthermore, our assembled sequence covered 96.44% of the transcriptome sequences (Supplementary Data 6). Together, the genome completeness as well as the contig N50 and scaffold N50 sizes improved considerably with respect to the previously published gayal and Indian mithun assembly[18,19]. The number of complete mammalian single-copy BUSCOs increased from 3,433

**Table 1 Comparison of *Bos frontalis* genome assembly statistics between this study and published gayal assemblies.**

|  | This study (Drung cattle) | Published by Wang[18] | Published by Mukherjee[19] |
|---|---|---|---|
| Sequencing method | Illumina + PacBio + 10×Genomics | Illumina | Illumina + PacBio |
| Assembly size | 2.74 Gb | 2.85 Gb | 3.00 Gb |
| Data volume (clean data) | 1228 Gb | 277 Gb | 250 Gb |
| Coverage (×) | 448 | 97 | 83 |
| Number of scaffolds (*n*) | 109,010 | 460,059 | 5015 |
| Scaffold N50 | 4.08 Mb | 2.74 Mb | 1.00 Mb |
| Longest scaffold | 35.74 Mb | 13.76 Mb | 6.54 Mb |
| Contig N50 | 157.67 kb | 14.41 kb | 11.50 kb |

(3,298) to 3,738 (Supplementary Data 5), the scaffold N50 size increased from 2.74 Mb (1.0 Mb) to 4.08 Mb and the contig N50 size rose from 14.41 kb (11.50 kb) to 157.67 kb (Table 1), corresponding to 8.88% (13.34%), ~1.5-fold (4.1-fold) and ~10.9-fold (13.7-fold) improvements, respectively. In comparison with the genome assembly of several bovine species, our assembled *B. frontalis* genome using the combined strategy showed relatively better quality in the Bovinae family (Supplementary Data 1).

**Whole-genome sequence anchoring.** To anchor the assembled scaffolds into pseudochromosome-scale sequences, the final Drung_*v*1.2 assembly was assigned and oriented on the *B. taurus* genome assembly. In the Drung cattle genome, 85.97% of sequences were anchored to the Btau_5.0.1 assembly, with 61.16% scaffolds covering 31 pseudochromosomes, including 29 autosomes and X and Y sex chromosomes, indicating that our assembled *B. frontalis* genome shared a high level of genomic collinearity with *B. taurus* (Fig. 1).

**Repeat structure analysis.** We identified a total of 1.23 Gb repeated sequences, which constituted nearly 44.92% of the assembled Drung cattle genome (Supplementary Table 1), similar to 43.69% in Mediterranean buffalo[14], higher than 37.21% in African buffalo[16], but less than that of taurine cattle (48.81%)[26] and wisent (47.03%)[11]. Among these sequences, 91.83% were transposable elements (TEs), accounting for 41.25% of the genome (Supplementary Data 7), which was lower than the published gayal (48.13%)[18] and a female Indian mithun (43.66%)[19]. The gayal genome has transposable element classes similar to those of other Bovidae families, as well as many ruminant-specific repeats. The most predominant repeat type (66.17%) of TEs was long interspersed nuclear elements (LINEs), of which 34.54% were LINE-RTE (BovB) and 30.34% were LINE-L1, followed by short interspersed nuclear elements (SINEs, 20.67%). In comparison to taurine cattle genome[8], the Drung cattle genome overrepresented ~40% LINE-L1 repeats and had ~87% more LINE-RTE (BovB) repeats.

**Genome annotation.** We used three gene-prediction methods (ab initio prediction, homology-based searching, and transcriptome sequence mapping) to annotate protein-coding genes in the Drung cattle genome and obtained 20,181 annotated protein-coding genes (Supplementary Data 8). Functional annotation showed that 20,003 predicted genes (99.12%) were searched within the available functional databases (Supplementary Table 2). Among them, a sum of 18,176 (90.06%) annotated protein-coding genes were expressed in at least one of the 14 tissues examined by RNA-seq (Supplementary Data 9 and Supplementary Data 10), supporting the high accuracy of gene annotation. Tissue-specific and housekeeping genes were also screened (Supplementary Data 11 and Supplementary Data 12). In all 14 tissues, 10,989 genes were shared, accounting for 60.46% of the total recorded and 54.45% of the total predicted protein-coding genes. Of these shared genes, 27 housekeeping genes were constitutively expressed ($\tau < 0.2$, based on the tissue specificity index[27]) (Supplementary Data 11). Additionally, we identified a total of 36,201 tRNAs, 1100 rRNAs, 1948 snRNAs, and 692 miRNAs in the Drung cattle genome (Supplementary Table 3).

**Genome comparison with closely related species.** To estimate Drung cattle speciation times and explore their genetic relationships, we constructed a phylogenetic tree based on 2301 single-copy gene families of six species in the Bovinae subfamily (*B. frontalis, B. taurus*[8], *B. indicus*[9], *B. grunniens*[12], *B. bubalis*[13], and *B. bison*[10]),

with the other five mammals (*Ovis aries*[28], *Capra hircus*[29], *Pantholops hodgsonii*[30], *Ceratotherium simum simum*[31], and *Homo sapiens*[32]) as the outgroup (Fig. 2a). The result showed that *Artiodactyla* diverged from the white rhinoceros of *Perissodactyla* 96.2 (76.7–110.3) Mya (million years ago). At approximately 21.4 (17.9–27.2) Mya, *Bovinae, Antilopinae,* and *Caprinae* in the *Bovidae* family shared a common ancestor of *Ruminantia*. Separation and diversification of bovids occurred 10.2 (7.5–13.7) Mya. In the six selected species of the tribe *Bovini*, the three genera (*Bos, Bison,* and *Bubalus*) differentiated into four clades, leading to (1) domestic cattle, that is, *B. taurus* and *B. indicus*, (2) *B. frontalis*, (3) *B. grunniens* and *B. bison*, (4) *B. bubalis*. Notably, we found Drung cattle diverged ~3.0 Mya, ~1.20 Mya earlier than the clade of domestic cattle, which indicated that *B. frontalis* was clearly distinct from *B. taurus* and splits off from the genus *Bos*.

We then performed comparative analyses of protein-coding genes among the genomes analyzed. The results showed that 6013 orthologous clusters were shared in the eleven mammals (Fig. 2b). Of the 20,181 protein-coding genes (8643 gene families), 143 gene families (915 genes) were specific to Drung cattle compared with the other ten genomes (Supplementary Table 4 and Supplementary Data 13). In addition, 228 gene families were expanded, while 240 gene families were contracted in Drung cattle (Fig. 2a). Among the four species of the genus *Bos*, Drung cattle showed more events of gene family expansion with a 50–100 increase in number. We also identified 1102 positively selected genes (PSGs) undergoing adaptive evolution that underlies the species diversification and habitat specificity of Drung cattle (Supplementary Data 14). Further functional characterization of the above genes demonstrated that genes unique to Drung cattle were enriched in three pathways: long-term depression (nervous system), vascular smooth muscle contraction (circulatory system), and regulation of actin cytoskeleton (cell motility). Genes exhibiting positive selection showed an increased involvement in almost all pathways related to genetic information processing and metabolism categories. Extensive expansion of genes occurred in tight junctions (cell communication) and cardiac muscle contraction (circulatory system). Genes subject to contraction had important functions in phototransduction (sensory system) and pathways associated with signal transduction (Fig. 2c and Supplementary Data 15).

**Segmental duplication.** We identified 62.65 Mb of duplicated fragment sequences that recently represented SDs ($\geq 1$ kb in length, $\geq 90\%$ sequence identity, Supplementary Table 5). These duplications accounted for 2.3% of the Drung cattle genome, which was lower than that in domestic cattle (3.1%) but higher than that in Tibetan antelope (0.8%). The enriched GO categories highlighted the genes associated with the sensory perception of smell, response to epidermal growth factor, and immune effector process (Supplementary Fig. 6). Moreover, olfactory transduction was the most prominent representative of the KEGG analysis (Supplementary Data 16), particularly in cases in which olfactory receptor (OR) genes were significantly enriched (268 of 297, 90.24%). The OR gene family is the largest multi-gene family in mammals and is responsible for detecting and discriminating between hundreds of odor molecules[33,34]. Among our 20,181 annotated genes in Drung cattle, we found 885 genes belonging to the OR supergene family.

**Characterization of the centromere regions in Rob (2;28) fusion chromosomes.** Karyotype and FISH analyses demonstrated that chromosomal 1 of *B. frontalis* (BFR1) was derived from the centric fusion of bovine Rob (2;28) between chromosomal 2 (BTA2)

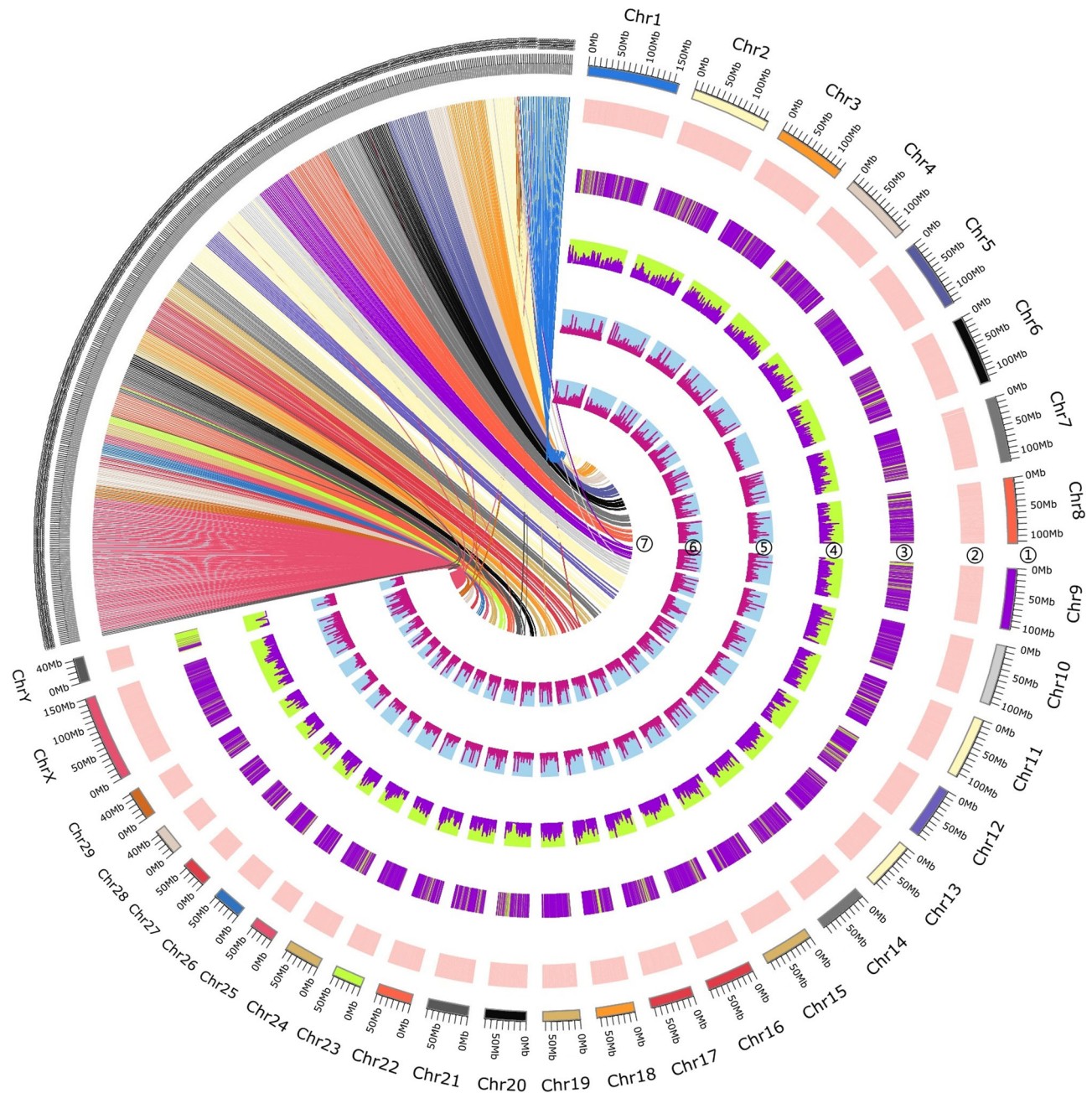

**Fig. 1 Overview of Drung cattle (*Bos frontalis*) reference genome.** Tracks from outer into inner circles indicate: ①The scale of chromosomes based on Btau_5.0.1; ②Synteny relationship of scaffolds between Drung_v1.2 and Btau_5.0.1 (all scaffolds/length >1000 bp); ③Gene density; ④Gene expression level; ⑤DNA methylation levels in heart tissue; ⑥DNA methylation levels in longissimus dorsi muscle; ⑦Synteny between *Bos frontalis* and *Bos taurus* genomes (scaffolds/length >100 kb, the lines representing syntenic regions colored by the chromosomes of *Bos taurus*).

and 28 (BTA28) of *B. taurus*[4,35]. Bovidae satellite I DNA (the 1.715 family) is believed to be a useful marker for the study of centromeric heterochromatin blocks in bovid chromosomes[36–38]. We found that there were concentrated tandem satellite I DNA repeats anchoring at the terminal end of the bovine autosomes, where 29 repeats were embedded within BTA2: 100,974,961–101,025,103 and 57 repeats were nested in BTA28: 39,416,521–39,507,756. We identified 29 scaffolds in our Drung cattle assembly containing homologous genes of BTA2 and BTA28 as the potential fusion regions of the Robertsonian chromosome. Among the 29 scaffolds, one scaffold (Fragscaffold60, part of BFR1) was detected to contain cattle satellite I with 28 tandem repeats at position 6,826,092–6,884,737, of which 16 genes were found to be aligned to BTA2 and BTA28 with at least

88.80% identity (average identity of 97.75%) (Supplementary Data 17).

Based on the alignment results, we propose a simulation model to speculate the mechanism of bovine Rob (2;28) involving the formation of Drung cattle chromosome 1 (Fig. 3a). Breakpoints occurred in the corresponding satellite sequences at the centromeres of BTA2 and BTA28, where chromosomal rearrangements and recombination facilitated centric fusion to form a large chromosome with a single, new centromere, namely, Drung cattle chromosome 1 (BFR1). Phylogenetic analysis of all extracted full-length satellite sequences showed that centromeric satellite I repeat in Fragscaffold60 clustered into an independent branch and formed a distinct satellite subfamily, whereas the

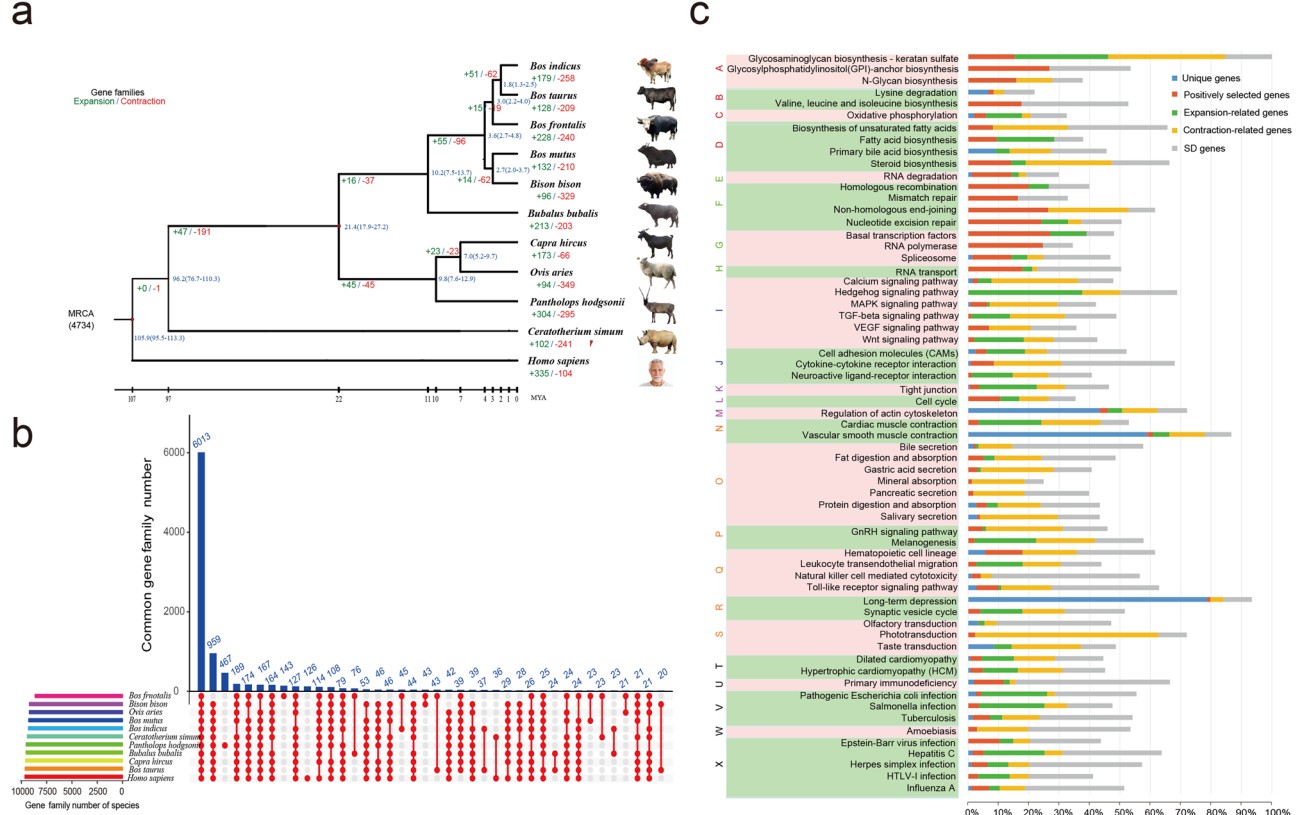

**Fig. 2 Evolution of gene clusters in Drung cattle (*Bos frontalis*) genome. a** Phylogenetic tree and dynamic evolution of orthologous gene clusters of *Bovinae* subfamily (*Bos frontalis, Bos taurus, Bos indicus, Bos mutus, Bubalus bubalis,* and *Bison bison*) in comparison with the selected five mammals (*Ovis aries, Capra hircus, Pantholops hodgsonii, Ceratotherium simum simum,* and *Homo sapiens*). Estimates of divergence time and its interval based on sequence identity are indicated at each node. The estimated numbers of orthologous groups (4734) in the most recent common ancestor (MRCA) are shown at the root node. The proportions and the number of orthologous groups that expanded (blue) or contracted (orange) in each lineage are shown on the corresponding branch (+, expansion; −, contraction). **b** Upset plot showing the shared and unique gene families among the eleven mammals. **c** Ratios and distributions of gayal-specific gene sets (unique genes, positively selected genes, expansion-related genes, contraction-related genes, and SD-related genes) in KEGG pathways. According to pathway maps, the gene sets are assigned to Metabolism (A–D), Genetic Information Processing (E–H), Environmental Information Processing (I, J), Cellular Processes (K–M), Organismal Systems (N–S), and Human Diseases (T–X).

repeats in BTA2 and BTA28 of *B. taurus* had high sequence similarity and tended to be classified into one category (Fig. 3b). These findings were consistent with the estimation analysis of divergence time (Fig. 2a), supporting the argument that Drung cattle are an independent lineage in the genus *Bos* and divergent from modern domestic cattle.

**Organization of centromeric satellite DNA (satDNA) in the centromere of BFR1.** It is worth pointing out that the organization of Drung cattle centromere satDNAs was comparable to that found in alpha satDNA of humans and primates, characterized by concurrent arrangements of monomers in simple or higher-order repeat (HOR) arrays. Tandemly arrayed satellite monomers were distributed in the centromere core of BTA2, BTA28, and Fragscaffold60 (Fig. 3c). We defined the HOR-like arrays as satellite blocks in the genus *Bos*. Here, two blocks on BTA2, eight blocks on BTA28, and three blocks on FragScaffold60 were identified, displaying approximately >85% DNA sequence identity, which indicated that centromeric satellites in cattle had chromosome-specific higher-order repeat units. Furthermore, each block array having a different length exhibited a plurality of repeat units with a 3–15% divergence between the constituents. Most remarkably, the two blocks (Block1 and Block3, inverted arrays) at the beginning and end of the centromere satellite core in Fragscaffold60 had the highest homology

with the corresponding starting (90.44%) and terminal (91.64%) inverted blocks at BTA28, respectively, when comparing the consensus sequences between blocks. Moreover, Block2 located in the middle of Fragscaffold60 had a closer sequence homology with the second (92.35%) of the two blocks in BTA2, presenting arrays in the same positive direction (Fig. 3c). This finding provided sequence evidence for recombination, supporting the view that the fusion of two ancestral chromosomes created a Robertsonian chromosome in Drung cattle. Specifically, the formation of recombinant centromeres in the evolution of Drung cattle chromosome 1 might be derived from two terminal satellite arrays of BTA2 and certain satellite repeats of BTA28.

**Genetic variation in centromeres.** The CENP-B box, a 17-bp recognition motif (YTTCGTTGGAARCGGGA) for the sequence-specific centromeric protein CENP-B, has been thought to be associated with functional centromeres[39,40] and linked to the HOR structure[41]. We evaluated the homology, distribution, and features of CENP-B boxes in satellite DNA sequences of Rob-related chromosomes using the consensus CENP-B box sequence containing nine core positions required to support DNA binding: NTTCGNNNNANNCGGGN[42,43]. Two motifs were similar to the consensus CENP-B box and identical in 5/6 out of 9 core nucleotides (Fig. 3d and Supplementary Fig. 7); in addition, the less strictly defined CENP-B element (AAACGGG)[44] was found

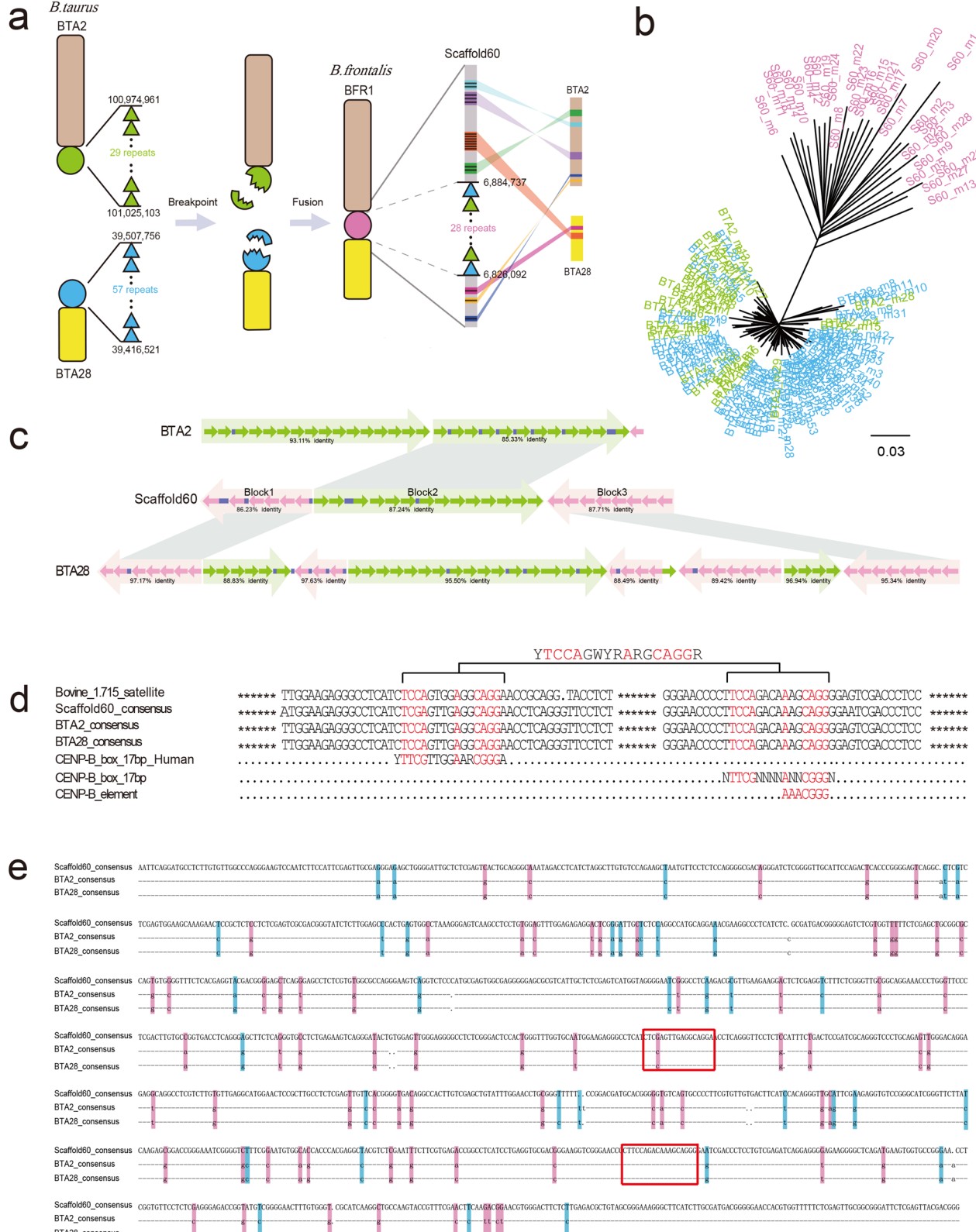

to have only one nucleotide transition (AAACAGG). Based on the identical sequences of the two alignments in each repeat unit, we propose an evolutionarily conserved motif (YTCCAGWYR-ARGCAGGR) for the putative CENP-B box in the genus *Bos*, in which three transitions (two G/A, one T/C) occurred in the nine core reignition binding nucleotides. In the full-length consensus satellite units, however, the rate of transversion (Tv, 67.62%) was

twice as high as the transition (Ts, 32.38%) among the 105 substitution mutations in Drung cattle compared with BTA2 or 28 (Fig. 3e). Moreover, the transversions tended to interchange between G and C/T, accounting for 70.42%. Sequence substitution saturation analysis showed that the transition rate increased linearly with genetic distance, whereas the trend of transversions was approximately saturated (Supplementary Fig. 8). Taken together,

**Fig. 3 Characterization of the structure of putative centromeric repeats in gayal chromosome 1 (BFR1) derived from Rob (2,28). a** Schematic illustration of centromeric fusion model for Rob (2,28) in *Bos frontalis*. BTA2 and BTA28 represent chromosomes 2 and 28 on *Bos taurus*, respectively. BFR1 indicates chromosome 1 on *Bos frontalis*. Scaffold60 is an abbreviation for Fragscaffold60, which is identified as a putative centromere-fusion scaffold in BFR1. The circle indicates the centromere region and the triangle indicates the satellite repeat. Considering the continuity of gene positions, we divide 17 syntenic genes, represented by black lines, into 7 alignment segments, where 5 segments with 9 genes in BTA2 and 2 segments with 8 genes in BTA28. **b** Phylogenetic analysis of the centromeric satellite I repeats in BTA2 (green), BTA28 (blue), and Scaffold60 (pink). **c** The mode of satellite repeat organization in centromere core characterized as the higher-order arrays. Based on high sequence identity in repeat monomer alignments, consecutive ordering and orientation, two blocks on BTA2, eight blocks on BTA28, and three blocks on Scaffold60 are identified in the centromere regions, respectively. Each block is composed of a multimeric satellite array representing a semitransparent arrow, called higher-order repeats (HORs) in humans and primates, in which the constituent monomers are distinguished (greater than 85% sequence identity). Solid arrows represent monomeric repeat units, with green and purple arrows depicting direct and inverted orientation respectively. Random insertions (purple rectangle) are found interspersed among the blocks. The illustrated syntenic patterns (gray parallelogram shading) describe the correspondences of the three satellite blocks in Scaffold60 with blocks in BTA2 and BTA28, respectively, having the highest homology in the alignment of the consensus sequences between blocks. **d** CENP-B box-like motifs extracted from consensus satellite sequences. The alignment of the CENP-B box sequences and its flanking regions are shown (Supplementary Fig. 7). Two motifs are comparable to the consensus CENP-B box respectively, where nine core nucleotides are denoted in red and the identical motif is "YTCCAGWYRARGCAGGR." **e** Nucleotide substitution of full-length satellite repeat consensus sequences. Transitions (Ts, 34/105) are represented in blue, and transversions (Tv, 71/105) are represented in pink. Dashes indicate identical bases and dots indicate deletion. Red rectangle shows the location of CENP-B box in satellites.

nucleotide substitutions in satellite DNA composition led to the diversity of satellites and made them valuable species-specific markers of phylogenetic relationships, while the functional motifs remained conserved to carry out the roles and responsibilities of the centromere.

**Network visualization of the Drung cattle gene expression profiles.** In an effort to explore comprehensive tissue-specific transcriptome profiling, 13,070 genes were selected to construct a gene co-expression network using weighted gene co-expression network analysis (WGCNA)[45]. A dynamic hierarchical tree algorithm divided the gene set into 29 co-expression modules (Supplementary Table 6 and Supplementary Fig. 9). Genes within a module were considered to share similar functions, and the stronger the correlation between the modules, the closer were their gene expression patterns (Supplementary Fig. 10). Under the criteria of $r > 0.90$ and $p < 0.001$, we identified nine functional modules that had corresponding unique tissues (Supplementary Data 18 and Supplementary Fig. 11). Notably, the high correlation coefficients between the magenta module and heart ($r = 1$, $p$ value $= 3.0 \times 10^{-15}$). All 372 genes in the magenta module were intensively annotated in circulatory system process, contractile fiber, and voltage-gated channel activity (Supplementary Fig. 12), and enriched in cardiac muscle contraction and some types of cardiomyopathies (Supplementary Fig. 13). The global network was visualized based on intramodular connectivity (Fig. 4a). We observed that some of the 29 modules exhibited their own distinct co-expressed gene clustering. Furthermore, hub genes with the highest intramodular connectivity in each module were discovered (Supplementary Data 18).

Out of the 18 hub genes identified in the heart-specific module (Fig. 4b and Supplementary Data 19), we noticed that three hub genes (*ANF*, *GABRB1*, and *LOC105001401*) were expressed exclusively in the heart (Supplementary Data 11). In the heart, the expression of ANF is associated with cardiac myocyte size[46], vascular smooth muscle cell proliferation[47], and endothelial cell growth and migration[48], which were important in angiogenic processes[49]. The *GABRB1* gene plays a role in mediating the fastest inhibitory synaptic transmission in the central nervous system, which was associated with the acute coronary syndrome and coronary heart disease[50] as well as body weight[51]. Furthermore, a biological interaction network map for gene prioritization and predicting the function of the 18 cardiac hub genes was constructed (Fig. 4c). These interacting genes are mainly involved in circulatory system and muscle system

processes, heart contraction, cardiac tissue morphogenesis, heart development, and actin cytoskeletal pathways. Among them, three core hub genes (*RYR2*, *TNNI3*, and *ACTC1*) had high degrees of connection and participated in more biological processes than the other hub genes.

**Genome-wide DNA methylation patterns.** DNA methylation is a form of chemical modification of DNA that affects its function, without altering DNA sequences[52]. To further explore the epigenetic characteristics of tissue-specific, we profiled the genome-wide DNA methylation status of four tissues (heart, lung, longissimus dorsi, and subcutaneous fat) for Drung cattle. The overall level of genome-scale DNA methylation in Drung cattle was approximately 3.11% (Fig. 1), ranging from 2.75% in the heart to 3.29% in the muscle (Supplementary Fig. 14a and Supplementary Data 20). Among the three DNA methylation contexts, the CG context was the primary contributor to DNA methylation, accounting for 96.87–97.39%, and mainly distributed in hypermethylated regions. In a parallel comparison, the frequency of hypermethylation sites in muscle tissue of Drung cattle was approximately 15% higher than that of *B. taurus* (Supplementary Fig. 14b). Furthermore, we divided the gene into specific gene features: gene body, transcription start site (TSS), transcription end site (TES), and the 2-kb region upstream of the TSS and downstream of the TES of each gene. For CG methylation profiling, we found that the four tissues exhibited the same trend in different gene elements. The methylation level showed a downward trend from upstream of the TSS to the TSS and then gradually increased, stabilized in the genomic region, and decreased slightly downstream of the TES (Supplementary Fig. 14c). In contrast, the methylation levels in the CHG and CHH contexts were generally constant throughout the genome, except for an obvious peak in the gene body region, with the lowest level in the lung and the highest level in the heart.

**Genes involved in the biological processes associated with the circulation system and myocardial contraction.** To explore the molecular genetic basis of cardiac adaptive evolution, we focused on the pathways (KEGG database: map04260, map04270, map04370, and map04810) of myocardial and vascular smooth muscle contraction related to the circulatory system. Then, we integrated a circulation and contraction-related biological process (Fig. 5a), from which 43 evolutionary genes of the Drung cattle were obtained, including 22 expansion genes, 17 positive selection

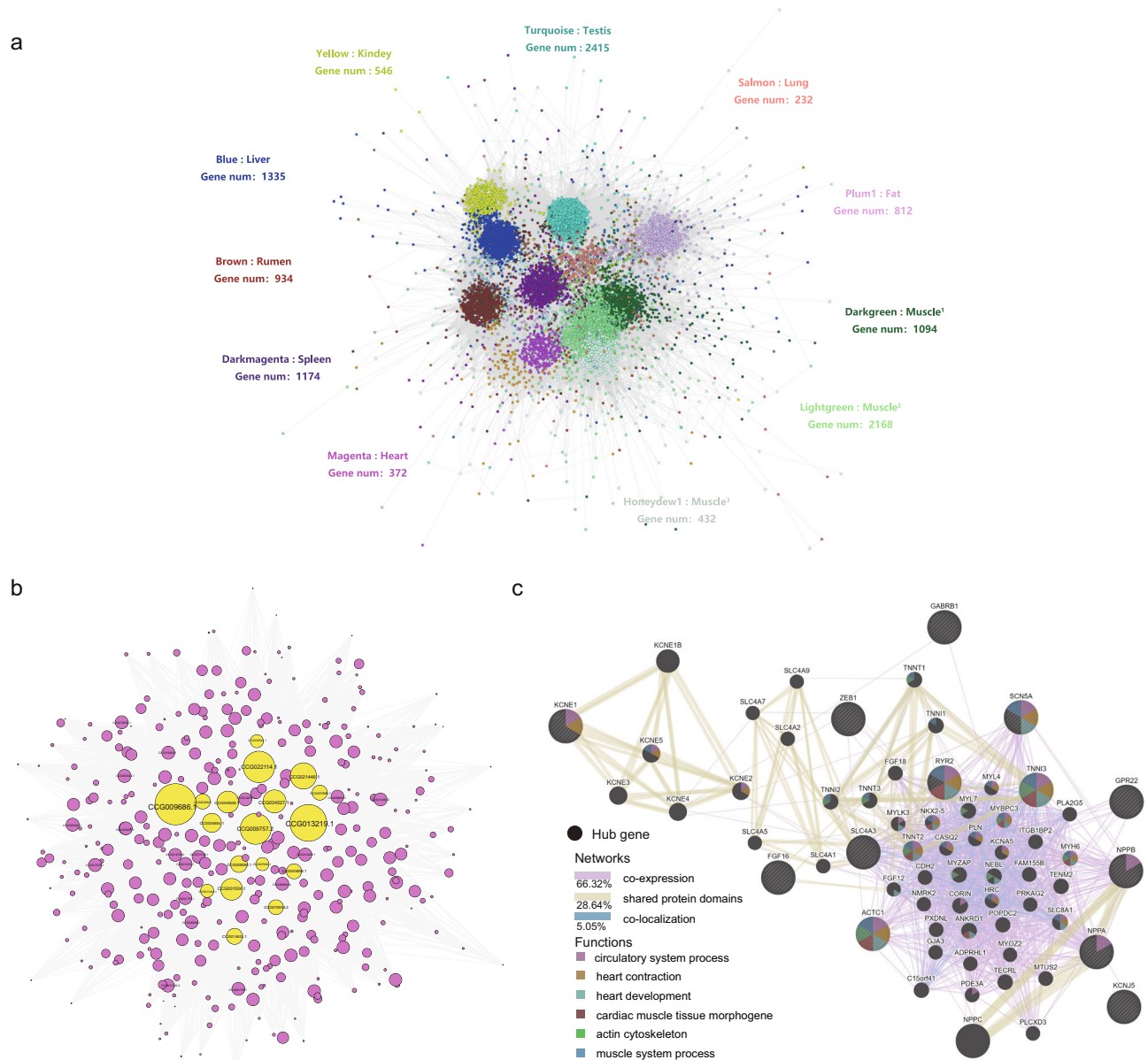

**Fig. 4 Network visualization and clustering of Drung cattle gene expression dataset. a** Gene module identified by weighted gene co-expression network analysis. Each node in the network represents a gene and the lines correspond to correlations between interacting genes. The colors indicate different modules. The functional module with the gene numbers and the corresponding tissue are indicated in the module color. Muscle[1] represents rump, tenderloin, chuck, and neck muscles; Muscle[2] represents striploin, rump, tenderloin, chuck, neck, and cardiac muscles; Muscle[3] represents striploin muscle. **b** Magenta-heart module. Hub genes were marked with yellow. The size of each node represents the degree of the intramodular connectivity of the node to adjacent genes. **c** The biological interaction network of 18 identified cardiac hub genes.

genes, three hub genes, and one Drung-specific gene (Supplementary Data 21). During the excitation-contraction coupling (ECC) process, the dihydropyridine receptor (DHPR) family, which mainly mediates $Ca^{2+}$ influx in excitable cells[53], showed an expansion of more copies in Drung cattle, possibly leading to improved efficiency of voltage-gated calcium channels. DHPR interacts with the underlying cardiac ryanodine receptor (RyR), mainly encoded by *RYR2* to initiate cardiac ECC[54]. The expansion of the DHPR family and cardiac-specific high expression of the *RYR2* gene may further enhance the storage and release of $Ca^{2+}$ in the sarcoplasmic reticulum (SR). Next, the growing influx of $Ca^{2+}$ could bind a greater amount of the troponin complex (TnC-TnI-TnT) to promote active involvement in the regulation of the actin-myosin interaction in the cardiomyocytes

forming cross-bridges. In particular, we identified TNNI3 encoding cardiac troponin I (cTnI) and ACTC1 encoding cardiac α-actin, with greater connectivity and higher expression levels as two of the three core hub genes in the heart function. Conversely, the increase in the number of $Ca^{2+}$ ions and the binding to calmodulin would strongly activate myosin light chain kinase (MLCK), in which the cardiac MLCK2 gene of Drung cattle was positively selected, thereby resulting in a cross-bridge formation leading to contraction[55]. Moreover, many vasoconstrictors could couple with G13 to activate the ROCK signaling pathway[56], consequently regulating myosin-mediated cell contraction. We found that the *ROCK* gene family was expanded in Drung cattle, suggesting that it may exert an effect on the advancement of cardiovascular function. Altogether, the combined effects of the

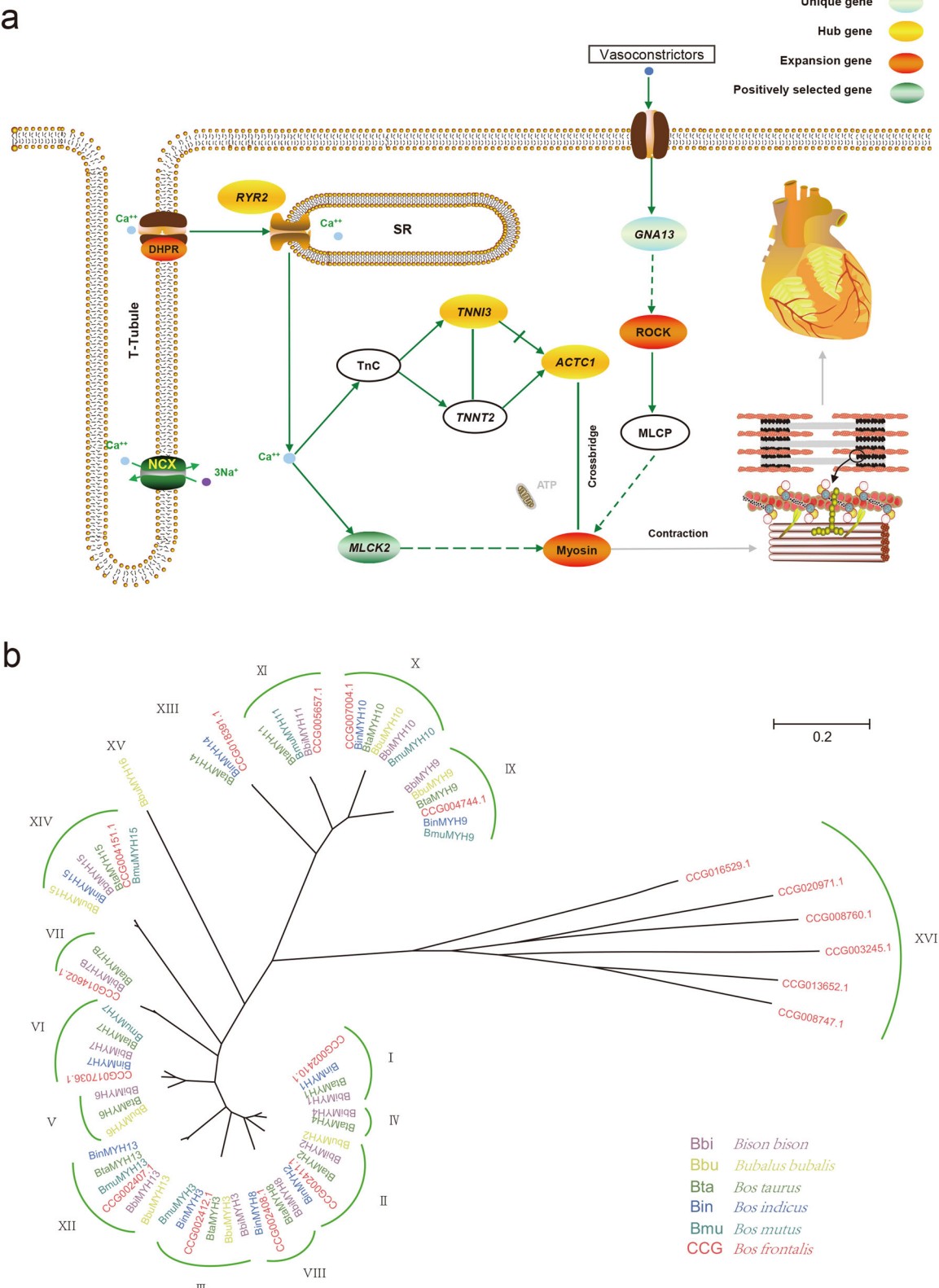

**Fig. 5 The evolutionary genes involved in the circulation and contraction-related biological process associated with Drung cattle adaption. a** The integrated circulation and contraction-related process. Unique gene, hub gene, expansion gene family, and positively selected gene are represented by aquamarine, yellow, red, and green ovals, respectively. Green solid lines indicate direct relationships. Green dashed lines indicate more than one step in the path. **b** Phylogenetic analysis of the *MYH* gene family in gayal and the other five Bovinae species. Each clade is assigned a Roman numeral (I–XVI).

expansion of DHPR and the ROCK family or the positive selection of the *MLCK2* gene, which evolved in the circulation and contraction-related biological process, ultimately contributed to the rapid evolution of myosin in Drung cattle.

Myosin is a very large and diverse superfamily of genes[57]. MYH is part of the components of Myosin II[58], which binds to actin and hydrolyzes adenosine triphosphate, providing force for diverse movements, such as cytokinesis, phagocytosis, and muscle contraction[59,60]. To further elaborate the evolutionary relationships of the MYH gene family, we compared amino acid sequences among six Bovinae species, including cattle, yak, Indian buffalo, zebu, American bison, and Drung cattle. In the phylogenetic tree of Bovinae's MYH gene family, a total of 16 branches were constructed, with gene members of similar functions clustering together (Fig. 5b and Supplementary Data 22). We found that all 15 *MYH* genes contained in the human MYH family could be discovered in all six selected species of *Bovinae*, corresponding to 15 clades; however, there were gene members missing in each species (Supplementary Data 23). Similar to zebu and yak, Drung cattle lost both *MYH4* (clade IV) and *MYH6* (clade V) during gene family evolution. As a pseudogene, *MYH16* gene was missing in the other five selected species except for buffalo. In addition, clade XVI with six gene members was only present in Drung cattle. Three of them belonged to unconventional myosin-I (myosin-Ib, myosin-Ic, and myosin-Ie), which might regulate membrane tension, cell adhesion, and actin architecture[61]. Overall, the adaptative molecular changes of evolutionary genes involved in the circulation and the contraction-related biological process might make an important contribution to the enhancement of cardiovascular system functions of Drung cattle.

## Discussion

As the wild relative of domestic cattle, the origin and evolutionary history of gayal have not been conclusively demonstrated. The specific karyotype ($2n = 58$) distinguishes gayal from gaur (*B. gaurus*; $2n = 56$) and cattle (*B. taurus* and *B. indicus*; $2n = 60$)[3,4]. The view that gayal is neither a domesticated type from wild gaur[62–64] nor a hybrid descendant from the crossing of wild gaur and domestic cattle[63,65], but an independent bovine species, is supported by Wu et al.[66], Walker et al.[67], and Ma et al.[68]. Research on the evolutionary relationship of gayal, gaur, and cattle is ongoing. The molecular phylogeny of the genus *Bos* inferred from the whole mitochondrial genome sequences showed that gayal (Drung cattle) clustered together with *B. indicus*, *B. taurus* as well as *Bos primigenius*, and had a close genetic relationship with wild aurochs than domestic cattle[69,70]. Baig et al.[71] and Chen et al.[72] demonstrate the close evolutionary proximity of *B. gaurus* and *B. frontalis* using mtDNA sequences. In contrast, Indian mithun was inclined to gather with gaur, suggesting that Indian mithun might be a descendant of gaur[73]. In our phylogenetic analysis, three genera (*Bos*, *Bison*, and *Bubalus*) were divided into four evolutionary clades after the separation of Bovini. This result indicated that Drung cattle was distinct from domestic cattle in lineage, which is consistent with the previous studies[18,65,69]. Excluding the earlier divergence of buffalo from bovids, it has been estimated that the separation of Drung cattle, yak, and bison occurred successively, approximately 3 Mya during the Late Pliocene Epoch (the Piacenzian Age, 3.6–2.6 Mya). In view of speciation times and genetic relationships in Bovina, Wu *et al.* demonstrated that genetic introgression has been pronounced from domestic cattle to wisent and yak, from zebu to gayal and banteng, in which introgression from zebu to gayal had a high level of population differentiation[66]. Molecular genetic analysis of populations from diverse geographic

locations will further advance understanding of Drung cattle evolutionary history.

Robertsonian translocations are usually considered the most predominant chromosomal rearrangements in the evolution of ruminant karyotypes[74–77]. Although the centric fusion of the Rob (2;28) translocation is known as the cause for the formation of chromosome 1 in Drung cattle[4,78], the precise mechanisms of centromeric DNA in fusion events and the contribution of satellite repeats to the centromeric structure remain unresolved. Cattle satellite I DNA (the 1.715 family), also named bovine SATI, is a 1.4-kb tandem repeat that constitutes the centromeric heterochromatin of all autosomes[36,38]. Here, the 1.715 satellite of bovine, found not only in *Ruminantia*[79,80] but also common in *Artiodactyla*[37], was used to characterize the centromeric DNA sequences in Bovidae[81,82]. One scaffold in our Drung cattle assembly showed sequence evidence of 28 tandem satellite I DNA repeats as well as 16 homologous genes shared with BTA2 and BTA28, which was inferred to be the centromeric fusion region of the Robertsonian chromosome through Rob (2;28). Next, we discovered satDNA arrangements with monomers or higher-order repeat arrays in the centromeres of cattle and Drung cattle, similar to human[83] and primates[84]. Given the sequence identity and divergence, the two inverted HOR-like arrays at the beginning and end of the identified Robertsonian fusion region in *B. frontalis* may be derived from the two terminals of BTA28 tandem satellite repeats, while its middle forward HOR-like array closely approximated one of the two satellite arrays of the BTA2 centromere. These observations facilitate an understanding of the formation of recombinant centromeres in *B. frontalis* as a result of a centric fusion translocation between homologous chromosomes 2 and 28 of the bovine ancestor. It is suggested that the evolutionary breakpoints occurring near or at centromeric satDNA homologous sequences promote chromosome rearrangements and contribute to karyotypic evolution in *B. frontalis*. Furthermore, the Rob translocation event has been deeply connected with the satDNA sequence homology and consequent recombination[85,86]. As reported by Sarah et al. for *Arabidopsis*, sequence substitutions led to the diversity of satellites in Drung cattle, which frequently occurred during centromere evolution. CENP-B protein (centromeric protein B) was highly conserved among mammals and recognized a 17-nt sequence known as the CENP-B box localized to centromeres[39,44,87]. Despite slight species-specific differences in binding site preferences, we noticed a consensus CENP-B box motif (YTCCAGWYRARGCAGGR) containing a 9 bp-core element that supported the conserved positions required for DNA binding in cattle and Drung cattle. Meštrović et al.[88] performed the CENP-B boxlike motifs analysis, confirmed the presence of a CENP-B boxlike motif in the monomers of this satDNA family, and allowed the occurrence of a maximum of four mismatches within the motif. It is generally recognized that Bovidae species exhibit high diversity in karyotypes, in which Robertsonian translocations undoubtedly promote the evolution of the Bovidae. Anchoring and localization of Rob (2;28) fusion, as well as characterization and organization of centromeric satellite repeats, will advance our understanding of the karyotype reorganization mechanism in *B. frontalis*.

Drung cattle are a rare and precious native cattle breed in China, with the dwindling population size and narrowing distribution range. Strong resistance to cold and humidity as well as high-stress tolerance to the environment are remarkable features of adaptability in Drung cattle. We first observed thick connective tissues as well as rich blood vessels and capillaries in the morphology of Drung cattle myocardial tissue, which might represent the phenotypic evolution reflecting the environmental adaptation of species to the forest-dwelling alpine valleys with torrid, humid, and rainy climates. Then, comprehensive analyses of comparative genomes, transcriptome

network, methylation, and cardiac histology provide insights into the adaptive evolution of Drung cattle. SDs have been considered the origins of gene gain, functional diversification, and gene family expansion[89–91]. Liu et .al. identified significantly enriched bovine SDs for specific biological functions such as immunity, digestion, and reproduction[92]. Our functional analysis of SDs in Drung cattle revealed the most significant enrichment in olfactory transduction ($p = 2.16 \times 10^{-35}$) related to odor sensation, in which 268 OR genes were found to have undergone segmental replication. In the integrated biological processes related to heart circulation and contraction, the key genes involved in the upstream and downstream pathways have changed in different evolutionary ways. In particular, three highly expressed core hub genes (*RYR2*, *TNNI3*, and *ACTC1*), two expanded gene families (*DHPR* and *ROCK* families), and one positively selected gene (*MLCK2*) constituted the framework for regulating biological pathways and ultimately promoted the expansion of the *MYH* gene family to meet the needs for the development and evolution of cardiovascular function. The *RYR2* gene has been reported to play an important role in the high-altitude adaptation of plateau-adapted animals, such as Tibetan wolves[93] and snub-nosed monkeys[94]. Expression of the *TNNI3* gene showed significant differences between affected and unaffected animals with bovine dilated cardiomyopathy[95]. The *ACTC1* gene, together with the regulatory protein tropomyosin and 3 troponins (C, I, T), was observed to form thin contractile filaments, representing the causative genes described in autosomal dominant heart failure[96]. Mutations in the *MLCK2* gene encoding myosin light chain kinase 2 have been linked to hypertrophic cardiomyopathy[97]. Along with the gene expansions of DHPR and ROCK on the two branch pathways, the combined effects facilitated the enhancement of the myosin function involved in cardiac contraction. Further phylogenetic analysis of the *MYH* gene family revealed that Drung cattle evolved species-specific gene members to adapt to the requirements of physiological functions. It needs to be stated that due to the scarcity in number and protection of Drung cattle in China, especially the need for slaughter, our transcriptome and methylome analysis were based on only one biological replicate, which may lead to some bias in our results. Nevertheless, our findings provide insights into the molecular mechanism underlying the environmental adaptability of Drung cattle.

## Methods

**Sample collection**. All samples were collected by trained personnel under strict veterinary rules. The animal component of this study was conducted in accordance with the Guidelines for the Experimental Animals, established by the Ministry of Science and Technology (Beijing, China). Animal experiments were approved by the Science Research Department (in charge of animal welfare issues) of the Institute of Animal Sciences, CAAS (Beijing, China) (IAS2020-96). Whole blood and 14 tissue samples (heart, lung, liver, spleen, kidney, rumen, testis, striploin, tenderloin, rump, chuck, neck muscle, the abdominal and subcutaneous fat) were collected from an adult male Drung cattle (*B. frontalis*) in the Drung Cattle Nature Reserve of Drung-Nu Autonomous County of Gongshan, Yunnan province, China. About 50 mL whole blood was collected from the jugular vein in a tube with EDTA, frozen at −20 °C, and stored at −80 °C. For each tissue, three samples were collected within 30 min postmortem, immediately frozen, and stored in liquid nitrogen (−196 °C). In addition, the muscle tissues were fixed in 10% buffered neutral formalin, routine histological processes were applied and tissue sections were stained with hematoxylin and eosin (H&E) to access the histomorphology analysis.

**Genome size estimation**. The flow cytometry (FCM) method was used to estimate the genome size of Drung cattle. Determination of nuclear DNA content by flow cytometry requires comparison with a reference standard. In general, to measure the DNA content of mammalian cells, chicken (*Gallus gallus domesticus*) erythrocytes (2 C = 2.5 pg) are commonly selected as the suitable internal DNA reference standard, with its genome size close to the sample and avoiding peak overlap. Here, lymphocyte cells from fresh blood of seven Drung cattle (*B. frontalis*) and two Simmental cattle (*B. taurus*) as control were isolated. Samples and chicken red blood cells were stained with propidium iodide (PI) and subsequently prepared for a standard flow cytometry procedure[98,99] using BD LSRFortesssa (Becton, Dickinson and Company, America). The genome size of each *B. frontalis* and *B. taurus* sample

was calculated by multiplying the cattle to chicken fluorescence ratio by the known value of the chicken genome size. Genome size can be calculated by counting *k*-mer frequency of the read data. To confirm the flow cytometry result, clean data of part insert paired-end libraries (200 bp, 450 bp, and 800 bp) was exacted to calculate the *k*-mer distribution using KmerGenie (*v1.7023*)[100] with default parameters to determine the best *k*-mer size. Compared to the traditional *k*-mer analysis, the hybrid model used by Kmergenie provides a better fit to the distribution of *k*-mer frequencies in complex genomes and modifies the deviation of the *k*-mer frequency from the Poisson distribution.

**Illumina paired-end library preparation and sequencing**. High-quality genomic DNA of Drung cattle was isolated from whole blood using the DNeasy Blood & Tissue Kit (Qiagen GmbH, Hilden, Germany), with RNase A (Qiagen, USA; 19101) digestion, following the procedure described. Genomic DNA (gDNA) was assessed for quality by NanoDrop® 2000 Spectrophotometer (Thermo Scientific, USA), and Qubit® fluorometer (Invitrogen, Carlsbad, CA).

According to the standard protocol of Illumina paired-end and mate-pair genome library construction, eleven Illumina WGS libraries were constructed with multiple insert sizes (200 bp, 450 bp, 800 bp, 2 kb, 5 kb, 10 kb, and 20 kb), and sequenced on Illumina® HiSeq X Ten or MiSeq platform.

**PacBio SMRTbell library construction and sequencing**. To generate long-read library, high molecular weight (HMW) blood genomic DNA (gDNA) was isolated using a standard phenol-chloroform extraction protocol and the AMPure XP beads (Beckman Coulter, USA) purification step. DNA quality control of purity, quantification, and concentration was measured by NanoDrop® 2000 spectrophotometer (Thermo Scientific, USA) and Qubit® fluorometer (Invitrogen, Carlsbad, CA). Subsequently, HMW gDNA was sheared to ~20 kb targeted size using ultrasonication (Covaris, Woburn, Massachusetts, USA). Size selection was made using a Blue Pippin instrument (Sage Science) according to the PacBio protocol "Procedure & Checklist – 20 kb Template Preparation Using BluePippin Size-Selection System." Library quality was assessed using the Sage Science™ Pippin Pulse Electrophoresis Power Supply System. Concentrate size-selected SMRTbell templates then proceeded to the DNA damage repair step with 1X AMPure PB Beads.

Single-molecule, real-time (SMRT) sequencing was performed on the Pacific Biosciences RS II instrument (Pacific Biosciences, CA, USA) using P6-C4 sequencing chemistry, with magnetic bead loading and 360-min movie lengths.

**10× Genomics library preparation and sequencing**. HMW gDNA was extracted from whole blood using the Qiagen MagAttract HMW Kit (Qiagen, 67563) referring to 10× Genomics sample preparation protocol "the CG00015_Sample-PrepDemonstratedProtocol_DNAExtractionfromBlood_RevB". Then, samples were analyzed via pulsed-field gel electrophoresis, and DNA fragments longer than 50 kb were used to construct one GemCode library using the Chromium instrument (10× Genomics, Pleasanton, CA). A subset of size-selected gDNA was processed with Chromium™ Genome Reagent Kits and sequenced on an Illumina HiSeq 2500 (Illumina, San Diego, CA).

**RNA isolation for transcriptome sequencing**. Immediately transfer frozen tissue into mortar, add liquid nitrogen, then crush with pestle to homogenize until powdery. RNA was extracted from 14 tissue samples using Trizol method, and RNA was subjected to quality control by the NanoDrop® 2000 (Thermo, CA, USA) and treated with DNase I (RNase-free) following the manufacturer's instructions. A total amount of 5 μg RNA per sample was used as input material for RNA-seq library preparation. Sequence libraries were generated using NEBNext Ultra™ RNA Library Prep Kit for Illumina (NEB, USA) and index codes were added to attribute sequences to each sample. To select cDNA fragments of preferentially 250–300 bp in length, the library fragments were purified with AMPure XP system (Beckman Coulter, Beverly, USA). After cluster generation, the library preparations were sequenced on an Illumina Hiseq 4000 platform.

**WGBS library preparation and sequencing**. Genomic DNA (1 μg) from each tissue sample (heart, lung, striploin, and subcutaneous fat) was sheared to a size of 200–300 bp. These DNA fragments were then used to construct a whole-genome bisulfite sequencing library following the manufacturer's instructions. Prior to sodium bisulfite mutagenesis, 0.5% unmethylated Lambda DNA was added as an internal control to monitor the bisulfite conversion rate. Briefly, the DNA fragments were processed by repairing 3′ ends, adenylating 3′ ends, and ligating adapters. The ligated DNA was converted with bisulfite using the EZ DNA Methylation Gold kit, Zymo Research (ZYMO). After processing, the methylated C became U (which became T after PCR amplification), while the methylated C remained unchanged. And then each sample of the WGBS library was obtained by PCR amplification. Finally, each WGBS library was sequenced using the Illumina Hiseq X ten platform.

**Genome assembly**. A total of 11 libraries with seven different insert sizes ranging from 200 bp to 20 kb were constructed. We also sequenced the library of 450 bp

insert size using Illumina MiSeq technology with a 2 × 250 bp read configuration in order to overcome the difficulty of assembling owing to abundant LINE-type transposable elements. DISCOVER de novo (v52488)[101] software was used to assemble the short reads into contigs and BESST (v1.0.4.4)[102,103] software was used to build these contigs into scaffolds. The gaps between contigs were filled with GapCloser (v1.12)[104], which assembled sequences iteratively in the gaps to fill large gaps utilizing the context and PE reads information. GapFiller (v1.11)[105] was applied to find reads that potentially fall within gaps and reliably close gaps within scaffolds by aligning paired reads. The primary contigs and scaffolds yielded assembly version Drung_v1.0, which comprised the genome size of 2.74 Gb with contig N50 of 18.17 kb and scaffold N50 of 1.68 Mb. The preliminary Illumina short-read-based assembly were then improved using PacBio and 10× Genomics Linked-Reads. In brief, the resulting Illumina contigs and PacBio reads (31X) were assembled with custom scripts (HABOT2, https://github.com/asarum/HABOT2) to fill gaps and form larger contigs based on seed-and-extend method. This Illumina-PacBio assembly, Drung_v1.1, had increased 8.6-fold and decreased 6.2-fold in size and number of contig N50, respectively. To order and orient the scaffolds into longer blocks, we used BWA (v0.7.15) mem to align the 10x Genomics data to the filled gaps assembly using default settings. The linked scaffolds were identified and oriented according to the patterns of reads from the same barcoded pools. The program fragScaff (v140324)[106] was used to perform scaffolding.

**Assessment of genome quality**. We first aligned the clean data of Illumina, Pacbio, and 10× Genomics to the final assembled Drung_v1.2 version genome, and obtained single base depth and alignment information for each locus. The base coverage, small fragment alignment, and base depth distribution were counted. Secondly, we employed BUSCO (https://busco.ezlab.org/) to access the content of single-copy orthologous detected in the Drung cattle genome. Thirdly, we used the transcriptome assembly contig (TAC) data to assess transcript coverage on the genome.

**Building pseudo-chromosomes**. To anchor the assembled scaffolds into pseudo-chromosome scale sequences, the final Drung_v1.2 assembly was assigned and oriented on the B. taurus genome assembly (Btau_5.0.1) using LASTZ (v1.02.00). AxtChain (v1.8.3) was used to chain together for alignments and chainPreNet (v1.26.0) removed un-nettable chains. Then we got the anchoring position of scaffolds corresponding to the chromosomes of B. taurus. The whole-genome syntenic relationship was visualized using Circos (v0.69)[107].

**Repeat identification**. Repeat sequences of the Drung cattle genome were identified with a combination of de novo and homolog-based methods. For the homolog-based methods, RepeatMasker (v4.0.9) and RepeatProteinMask (v4.0.3) were employed to find the transposons components against RepBase library with an e-value cutoff $1e^{-5}$ at DNA and protein level, respectively. Also, we used Piler (v1.0)[108] and RepeatModeler programmers to build the de novo gayal repeat library that was applied to find and classify the repeats. The identified repeat sequences were used to construct a non-redundant repeat sequence library, and repeat sequences with an identity of > 50% were grouped into the same classes. Tandem repeats were predicted by Tandem repeats finder (TRF) (v4.10.0)[109].

SDs were assessed with the assembly-dependent method and detected using LASTZ (v1.02)[110] with default parameters. The resulting alignment was filtered for maximum simultaneous gap ≤ 100 bp to identify fragments corresponding to recent segmental duplication (≥ 90% identity and ≥ 1 kb in length)[92].

**Gene prediction and annotation**. We integrated ab initio gene prediction, homologous sequence searching and transcriptome sequence mapping to conduct gene prediction of the B. frontalis genome. First, GlimmerHMM (v1.1.0)[111], genescan (v1.0)[112] and Augustus (v3.0.2)[113] were used to carry out de novo prediction. We downloaded B. taurus, C. simum simum, B. grunniens, B. bison, and B. bubalis from NCBI (https://www.ncbi.nlm.nih.gov/) and then aligned these protein sequences to B. frontalis using TBLASTN (v2.2.25)[114] for homology-based gene prediction. Annotation of the predicted genes was performed by combining InterPro[115], Gene Ontology[116], KEGG[117], and Swissprot with TREMBL[118] database. The noncoding RNAs were identified by searching the genome assembly against the Rfam11.0[119] database. tRNA was identified by using tRNAscan-SE (v1.21)[120].

**Anchoring and localization of satellite DNA in the centromere regions**. The BLAST-Like Alignment Tool (BLAT) (v3.2.1)[121,122] was used to match annotated coding sequence (CDS) of B. frontalis against Btau_5.0.1. Scaffolds in which genes can be aligned to BTA2 and BTA28 of B. taurus were identified as putative chromosomal fusion regions. The Bovidae satellite I DNA (1.715 satellite, GenBank: J00036.1) is a 1.4-kb tandem repeat[123,124] located in the centromeric regions of all 29 bovine acrocentric chromosomes[5,125]. We searched the sequence of Bovidae satellite I within Drung cattle putative chromosomic fusion scaffolds as well as BTA2 and BTA28 of B. Taurus using BLASTN (v2.2.25), respectively. All full-length cattle satellite I sequences were extracted to make multiple sequence alignments by ClustalX (v2.1)[126]. For analysis of the centromeric protein CENP-B,

the CENP-B box-like motif (NTTCGNNNNANNCGGGN)[42,43] and CENP-B element (AAACGGG)[44] were searched in the extracted satellite sequences using MegAlign (v5.05) to identify evolutionarily conserved sites exclusive to bovine. For analysis of sequence variation in satellite DNA repeats, nucleotide variation, base substitution rates, and p-distance were further conducted with MEGA (v7.0)[127] to assess the sequence substitution saturation.

**Phylogenetic tree construction and speciation time estimation**. We selected six species in the Bovinae subfamily (B. frontalis, B. taurus[8], B. indicus[9], B. grunniens[12], B. bubalis[13], and B. bison[10]) and other five mammals (O. aries[28], C. hircus[29], P. hodgsonii[30], C. simum simum[31], and Homo sapiens[32]) as the outgroup for phylogenetic analysis of the B. frontalis. The genes and protein-coding sequences of the ten mammalian species downloaded from NCBI and our B. frontalis assembly genome were used to define gene families. TreeFam (v9)[128] clustered all the genes that were grouped into orthology/paralogy and provided the evolutionary history of genes. The phylogenetic trees were constructed based on single-copy gene families by the PhyML (v3.0)[129] software. Calibration time was obtained from the TimeTree database[130].

**Gene family and positive selection analysis**. The protein-coding gene set of the eleven mammalian species was downloaded from NCBI and used to identify gene families that descended from a single ancestral gene in the last common ancestor of the given species. Cluster analysis of gene families was performed by TreeFam (v9)[128]. CAFE (v3.0)[131] was employed to obtain the rate of expansion and contraction of gene family under a random birth and death model. PAML (v4.9)[132] was employed to detect genes under positive selection for each single-copy orthologous gene family. For branch-site model analysis, we used all single-copy orthologous gene regions of eleven mammals and selected Drung cattle in the foreground position on the phylogeny. We then compared the result of null model (omega = 1, NSsites = 2, omega = 1 and fix_omega = 1) with the alternative model (omega = 1.5, NSsites = 2, omege = 1.5 and fix_omega = 0) using likelihood rate statistics method of CODEML software for each branch-site model of every orthologous gene. $\chi^2$ distribution was used to evaluate the significance ($p < 0.05$) of the likelihood ratio statistic.

**mRNA data analysis**. We used FastQC (http://www.bioinformatics.babraham.ac.uk/projects/fastqc/) to check and visualize the quality of raw data. First, we adopted relatively stringent criteria for quality control by removing reads containing adapters, ploy-N or low quality from raw data. The clean reads were then mapped to the B. frontalis genome and the Btau_5.0.1 assembly using Bowtie2 (v2.2.3)[133] and TopHat (v2.0.12)[134] with the default parameter settings, respectively. The transcripts were quantified with fragments per kilobase of exon per million fragments mapped (FPKM) of each sample by Cufflinks (v2.2.1)[134].

**Weighted gene co-expression network analysis**. The gene co-expression modules and network were constructed using WGCNA (v1.41) package[45]. RNA-seq reads were generated from 14 tissues, including seven organs (heart, lung, liver, spleen, kidney, rumen, testis), five muscles (striploin, rump, tenderloin, chuck, neck), and two adipose tissues (abdominal fat, subcutaneous fat). Genes with FPKM < 5 in 14 tissues were filtered out. The module was identified using the dynamic branch cutting method. Tissue-associated modules were selected with a correlation coefficient > 0.90 and Fisher's exact test p values < 0.01[135]. We constructed a global co-expression network of 29 identified modules using Cytoscape (v3.0)[136,137]. The top 5% of the genes with the highest connectivity in the network were designated hub genes[138,139]. An interaction network was built using the GeneMANIA database (http://genemania.org).

**WGBS data analysis**. Raw bisulfite sequencing data were filtered using Trimmomatic[140] software by removing low-quality reads. The filtration criteria were as follows: (1) Remove the adapter sequences; (2) Cut off bases or N (non-AGCT) with base quality values less than 3 on both sides; (3) Slide the window with 7 bases, and then cut off the average base quality values < 15 in the window; (4) Remove the reads with a length less than 75 bp or the average of base quality value less than 20. The clean data were aligned to our Drung_v1.2 genome using Bismark (v0.12.2)[141] with the default parameters. The degree of bisulfite conversion was estimated by internal phage lambda DNA control and found to be > 99%. The methylated cytosines (mCs) covered by at least five reads were used to calculate the methylation level. The methylation level was determined by dividing the number of reads covering each mC by the total reads covering that cytosine (the number of mC reads and the number of nonmethylation reads). We then assessed the methylation status at the whole-genome level and gene element level. Methylation levels > 75% were identified as the hypermethylated regions[142]. We analyzed the relationship between three DNA methylation contexts (mCG, mCHG, mCHH, where H stands for all nucleotides except guanine) and sequence preferences in nine base pairs (bp) spanning methylcytosine sites. To investigate the dynamic changes in DNA methylation, the distributions of methylation levels 2 kb upstream of the TSS and gene body regions and 2 kb downstream of the TES were compared.

**Statistics and reproducibility**. All Illumina paired-end, PacBio SMRTbell, 10× Genomics, RNA-Seq, and WGBS libraries included technical replicates within and among libraries. Statistical analyses were conducted using the cited packages and reproducibility can be achieved using the parameters reported in the Methods.

**Reporting summary**. Further information on research design is available in the Nature Research Reporting Summary linked to this article.

## Data availability

Sequence reads of the Drung cattle genome project have been deposited at the National Genomics Data Center (NGDC) (https://bigd.big.ac.cn/). Sequencing data of the Drung cattle genome assembly using Illumina, PacBio, and 10× Genomics platforms is available via accession PRJCA004132. The accession numbers of sequencing data of transcriptome and DNA methylation are PRJCA002143 and PRJCA003336, respectively. The source values for each of the following figures are available in the corresponding Supplementary Data files: Fig. 1—Supplementary Data 10, Supplementary Data 20; Fig. 2—Supplementary Data 13–16; Fig. 3—Supplementary Data 17; Fig. 4—Supplementary Data 12; Fig. 5—Supplementary Data 21–22. All other data are available from the corresponding author on reasonable request.

## Code availability

We used the following software in the analysis: R (v3.6.1), DISCOVER (v 52488), BESST (v1.0.4.4), GapCloser (v1.12), GapFiller (v1.11), BWA (v0.7.15), fragScaff (v140324), KmerGenie (v1.7023), Bowtie (v2.2.3), TopHat (v2.0.12), Cufflinks (v2.2.1), Piler (v1.0), Repeatmasker (v4.0.9), TRF (v4.10.0), LASTZ (v1.02), GlimmerHMM (v1.1.0), genescan (v1.0), Augustus (v3.0.2), TBLASTN (v2.2.25), tRNAscan-SE (v1.21), AxtChain (v1.8.3), chainPreNet (v1.26.0), Circos (v0.69), ClustalX (v2.1), MEGA (v7.0), TreeFam (v9), PhyML (v3.0), CAFÉ(v3.0), PAML (v4.9), WGCNA (v1.41), Cytoscape (v3.0), Bismark (v0.12.2). All codes can be made available upon reasonable request.

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

## Acknowledgements

This study was supported by the National Natural Science Foundation of China (31572376) and the Agricultural Science and Technology Innovation Program in the Chinese Academy of Agricultural Sciences (ASTIP-IAS03).

## Author contributions

X.G., Y.C., and J.L. conceived and designed the project. S.X. coordinated and arranged sample collection preparation. B.S., G.Y., L.S., and W.X. provided and collected blood and tissue samples of the Drung cattle. M.X. and R.Z. conducted flow cytometry experiments and genome size analysis. W.Y. and M.X. performed the genome assembly and annotation. Y.C., W.Y., and T.Z. performed evolutionary analyses. Y.C., X.G., R.Z., and M.X. carried out the analyses of repeat sequence and chromosome fusion. T.Z., L.X., and H.G. implemented transcriptome, methylation sequencing, and analysis. Y.C., X.G., and T.Z. performed the analysis of environmental adaptability. T.Z. and W.Y. carried out data submission and database construction. Y.C. and T.Z. wrote the paper. X.G. and J.L. revised the paper.

## Competing interests

The authors declare no competing interests.
