## [Peer Review File · Communications Biology]

Reviewers' comments:

Reviewer #1 (Remarks to the Author):

In the paper entitled "A draft genome of Drung cattle (*Bos frontalis*) reveals clues to its chromosomal fusion and environmental adaptation" by Chen et al., the authors assembled a better reference genome of Chinese specific Drung cattle, which can be a good genetic resource of further genetic research and conservation of the Drung cattle. Most importantly, this research revealed the formation mechanism of Robertsonian chromosome in *Bos frontalis*. The overall description of the article is rigorous and explicit. Below is a few issues and concerns that can make this manuscript better and more applicable to be published on Communications Biology.

Major issues:

1. line 62: the reference of domesticated yak genome, refer to the newly published version by Zhang et al. doi: 10.1093/molbev/msab134; Importantly, the authors should note that *Bos mutus* is wild yak and *Bos grunniens* is domestic yak, you misused it broadly in the paper. If needing to referring to the genome of wild yak, see Liu et al., doi: 10.1038/s41597-020-0400-3.
2. line 75; Figure S3: Did the result conduct by authors? I didn't see any linked method description. If not, add reference to original research.
3. Figure 1.2: The description is ambiguous; the blocks seem to be all the same.
4. lines 249-255: As the authors described, I suppose that Fragscaffold60 seemed to be partial BFR1 that contains the centromeric region? The authors need to add statement on this as ones may think that scaffold60 is BFR1.
5. Below are a few concerns on the result of chromosome fusion in Drung cattle.
 - 1) Assembled sequences in centromeric region always contains gaps because its property of highly repetitive. How about Fragscaffold60 in Drung_v1.2?
 - 2) There were only three copies of repetitive blocks. Except for Fragscaffold 60, did any other scaffolds contain similar blocks? Was the copy of CNEP-B box contracted in Drung cattle and how's the affect to chromosome stability?
 - 3) How was the order and orientation of genes of Drung_v1.2 on the scaffold60 comparing to their homologous ones on the genome of bovine? This is worth verifying to further support the centromeric fusion model.
6. Below are a few consents on the result of the biological process.
 - 1) line 434: expansion is a relative conception. I suggest authors to list the related gene families containing expanded genes and compare the number of genes clustering in which across all mentioned mammals.
 - 2) line 479: genes in the clade XVI seems have massive genetic variations with genes in other clades. Are they under positive selection in Drung cattle? If they share nonsynonymous mutations comparing to others, they may take important role in adaptation in Drung cattle.
 - 3) Are there similar researches on the expansion genes that the authors mentioned in the paper, for example, the copy number affect the function.
7. lines 494-504: For the domestication and phylogeny of *Bos* genus species, Zhang et al., doi: 10.1111/age.12974, presented a comprehensive review, the authors may refer to it.

Minor issues:

1. line 51: year number is comma-delimited, while some numbers were not comma-delimited in supp. note.
2. lines 148-156: need a space between LINE-RTE and (BovB). There were many similar issues in the main text and supp. note, i.e. space between text and left parenthesis.
3. line 1325 Figure 2b, it is an upset plot, not venn plot.
4. line 1386: add the label to y-axis in Figure 5e.
5. In Method and Supplementary Note section, the authors should add the version of all software and tools.
6. line 185 and line 217: use consistent expression of e-value cutoff.
7. lines 228-229: Add the resources or references to these genome sequences.

Reviewer #2 (Remarks to the Author):

The authors performed the de novo genome sequencing of a Drung bull by combining HiSeq&MiSeq, PacBio long-read sequencing, and 10x Genomics platforms to generate an improved *Bos frontalis* genome assembly. To evaluate the quality of a new assembly, several methods have been used, which include contig and scaffold size measurements (contig N50, scaffold N50), gene set completeness, synteny comparison, and the Benchmarking Universal Single-Copy Orthologs (BUSCO). Next, they performed several analyses to better understand Drung cattle origin and its taxonomic status, gene duplication, and biological process of local adaptation, and the mechanism of chromosomal fusion at the genome-scale. Finally, a combined analysis of transcriptomics and DNA methylation of Drung cattle was carried out to identify functional genes and understand the molecular biological mechanisms associated with animal adaptation in the challenging ecology of alpine valleys.

Overall, they used high-level technology to sequence the genome of gayal and they reported improvement in genome assembly which is good. However, many comparisons are made using the methylation and transcriptome data but most of them lack a proper assessment of statistical significance as only one biological replicate has been used. This is a strong limitation to draw reliable conclusions from the tissue comparison. Also, at several occurrences the methods should be clarified, better documented, and cleared of potential biases. In the introduction section, there is a bit of historical information and would benefit from focusing more on problems of existing genome assembly for gayal.

My review is divided into two parts: (A) Major general issues with detailed comments, and (B) Minor issues.

A) Major issues

1) Introduction needs to be revised completely. Much effort has been put into describing the habitat and history of the Drung cattle while focusing less on problems of existing genome assembly. It is necessary to explain factors affecting the accuracy and continuity of the genome assembly and how the current study is supposed to address those problems.

2) Method section writing should be improved. The writing should be improved to better explain the methods. Several steps of the analysis are not clear or insufficiently documented. It is not clear how sequencing, library preparation, and quality control have been done for genome and transcriptome sequencing and methylation calling. It is necessary to know the coverage and the number of reads for different tissues. Several statistical tests have been performed without providing enough details about statistical tests, significance levels (p-value), etc. It is not clear which database has been used for downloading genes and protein sequences for each species. In the gene ontology analysis, it is not clear what background set has been used. The version of all used software has not been documented.

3) Incompatibility between results of the phylogenetic tree and previous speciation studies. The authors estimated the divergence time based on single-copy gene families and reported the divergence time of *Bos taurus* from *Bos indicus* at 1.8 Mya (1.3-25). They also reported that *Bos frontalis* evolved ~1.20 Mya earlier than domestic cattle. However, these estimates are a bit far away from previous estimates. For example, a previous study (Wang et al 2017) estimated the divergence time of gayal from cattle and zebu at approximately 5.1 million years ago. Also, previous studies using mitochondrial genome, microsatellite, etc reported divergent times of *Bos taurus* from *Bos indicus* less than 1 million years. The authors should discuss how confident are their results? How much is the resolution of estimates? Compare current results with previous studies.

4) Lack of significance for transcriptomic and methylation study. Only one biological replicate is included for all tissues analyzed in the study. This limits the proper conclusion from tissue comparison and the author needs to take caution in this regard. For example, in abstract lines 27 the authors have reported the lowest methylation for the heart. How did you conclude this with only one biological replicate per tissue? How did you calculate the P-value? This limitation ought to be indicated somewhere in the manuscript though. I think it would be great if the author could validate their results.

B) Minor issues

P.3, l.12- "Bosin" => Bison

P.3, l.13- "MYa" all abbreviation should be explained at first => Million years ago (MYa)

P.3, l.14- "suggesting", => "confirming .."

P.3, l.16- "We found a consensus conserved CENP-B...", I would suggest saying "we confirmed", as a previous study had been reported it in the Bovidae. (Escudeiro, A., Adegas, F., Robinson, T.J., Heslop-Harrison, J.S., and Chaves, R. (2019) Conservation, divergence, and Functions of Centromeric Satellite DNA Families in the Bovidae. *Genome Biology and Evolution*. 11 (4), 1152–1165)

p.5, l.51- "1,986" => 1986

p.5, l.66-68- "In comparison with the above-mentioned bovine species, the two published genome assemblies of *Bos frontalis* had a relatively lower completeness." It is better to explain in more detail.

p.6, l.74-78- "Compared to the other native cattle in Yunnan, the muscle fibers of Drung cattle are smaller in diameter, higher in density and larger in number capillaries". I didn't find anything in the method section about these comparisons. Did the author did this by themselves or this is from another study, so please provide the reference of the study or document it in the method section.

P.6, l.82- "2n=58" => 2n=58 karyotype

p.6, l.95- "3-year-old male Drung cattle" => 3-year-old Drung bull

p.7, l.104- "10XGenomics" => 10x Genomics

p.8, l.133- remove the link

p.9, l.172-173- How robust is your gene set considering only one biological replicate you have? what is the tissue specificity index? Please explain how was calculated?

p.10, l.205-208- "A further functional characterization ... and regulation of actin cytoskeleton (Cell motility)." Please provide P-value for enrichment results.

p.11, l.223-228- "The enriched GO categories (Supplementary Fig. 7) ... genes were significantly enriched (268 of 297, 90.24%)." Please report all related P-values!

p.16, l.377-378- "had high degrees of connection, and participated in more biological processes than other hub genes" please report significant tests and P-values?

p.18, l.436- "DHPR family". Stand for what? Please provide full names for all gene abbreviations throughout the manuscript.

p.22, l.563-566- please reference a previous study that discovered the CENP box in cattle before your study and discusses it. "Escudeiro, A., Adegas, F., Robinson, T.J., Heslop-Harrison, J.S., and Chaves, R. (2019) Conservation, divergence, and Functions of Centromeric Satellite DNA Families in the Bovidae. *Genome Biology and Evolution*. 11 (4), 1152–1165"

p.23, l.584-588- "Our functional analysis of SDs in Drung cattle revealed the most significant enrichment in genes were specific to Drung cattle." Please provide p-value and explain which kind of enrichment you did (BP, CP, MP,)? explain what is the background gene?

p.24, l.610-612- Among the identified tissue-specific modules, the heart-specific module presented a significant co-expressed gene clustering," provide statistical test and p-value within a parenthesis.

p.24, l.616-618- "The whole genome DNA methylation profiling showed the lowest level in the heart with more non-CG DNA methylation compared to other tissues." please provide statistical test and p-value within a parenthesis. How significant and precise is this considering you used only one biological rep.

p.26, l.660- "Genome sequencing and assembly" please provide more information about the way library preparation, sequencing, and quality control have been done!

p.27, l.689- "Transcriptome sequencing" please provide more information about the way library preparation, sequencing, and quality control has been done!

p.27, l.694-696- "Clean reads were mapped to the *Bos frontalis* genome and Btau_5.0.1 assembly using Bowtie136 and TopHat137, respectively." Why two mappers have been used? Did you use the default parameter for mapping with each?

p.27, l.712-714- "We then aligned protein sequences of *Bos taurus*, *Ceratotherium simum simum*, *Bos mutus*, *Bison bison* and *Bubalus bubalis* to *Bos frontalis* by using TBLASTN145 for homology-

based gene prediction." Please provide a reference to the database for each species protein sequence you downloaded.

p.27, l.714-716- "Annotation of the predicted genes was performed by blasting the sequences against InterPro146, Gene Ontology147, KEGG148, Swissprot and TREMBL149 database." Please rewrite the sentences. To me, it seems you blast sequence against all databases!! while Gene ontology analysis uses gene lists not sequence as input!

p.28, l.743-744- "The protein-coding genes of the ten mammalian species downloaded from NCBI". Did you use RefSeq or Ensembl? What is the release date? etc Please give detailed information.

p.29, l.750-752- "The protein-coding genes set of the eleven mammalian species were used to identify gene families that descended from a single ancestral gene in the last common ancestor of the given species.". the same as my previous comment. Please reference all databases used.

p.29, l.756- "dN, dS and dN/dS." => please add explanation what is dN, etc?

p.29, l.758- "FPKM" => please add full name

p.29, l.769-770- "WGBS libraries were constructed according to the manufacturer's protocol." please provide more information about extraction, sequencing, methylation calling software used and parameters.

p.29, l.772- "The methylation level" What is the methylation level? How has been calculated?

p.29, l.774- "mCG, mCHG, and mCHH" stand for what? Explain in the main text.

p.29, l.776-777 How TSSs and TES were defined? please document which annotation data you used?

Figures- Generally, fonts are unreadable and need to be increased.

Fig2. To me, pie charts are not informative. What are you trying to add to the figures by adding pie charts? I would suggest omitting them to simplify the figure.

Reviewer #3 (Remarks to the Author):

Brief summary of the manuscript

The manuscript by Chen and Zhang et al. generates an impressive amount of genome, transcriptome, and methylome data analyzed to identify unique biological aspects of Drung cattle. A detailed summary of data generated and analysis conducted are enumerated below.

Summary of data generated:

Chen and Zhang et al. generate short-read Illumina sequencing, long-read PacBio sequencing, and linked-read 10XGenomics data. These data are assembled to a relatively high quality (demonstrated by measures such as N50, BUSCO score, repeat content) genome assembly of Drung cattle (*Bos frontalis*).

Genome annotation is done using ab initio prediction, homology-based approaches, and primary transcriptome data (14 tissue samples) generated in this study.

Genome-wide methylation levels of 4 tissues are generated. The methylation level is estimated using Bismark software to obtain the landscape of genome-wide DNA methylation.

Summary of analysis conducted:

 The genome assembly is compared with genomes of other species to identify changes (expansion/contraction) in gene content (CAFE analysis, upset plot, and GO enrichment analysis).

 The putative location of the Robertsonian fusion of BTA2 and BTA28 is identified. Gene synteny analysis, repeat annotation, and pairwise genome alignments analysis are used to try and reconstruct the sequence of events involved in this karyotype change.

 Gene co-expression networks are identified using the WGCNA approach.
 Changes in gene content (prominent examples being the loss of MYH4 (clade IV) and MYH6 (clade V)), the prevalence of positive selection, and changes in gene expression patterns are identified in genes of the circulation and contraction-related biological processes. These genes are proposed to be involved in the adaptation of Drung cattle.

Overall impression of the work

The manuscript is a significant contribution to further our understanding of the genomics and basic biology of *Bos frontalis* (Mithun/Gayal/Drung cattle). The datasets generated in this study will be of great value. The text is easy to follow and well written (see minor comments for a list of suggested changes).

Authors should provide more details (see major comments) regarding the methods used in the manuscript to support their novel results. Some additional analysis to support the significant claims also seems to be required.

Few lacunae in the approaches used and missing references are listed in the comments below.

Specific comments, with recommendations for addressing each comment

Major comments:

Comment 1: The authors need to state if, according to them, the Chinese gayal, Indian Mithun, Dulong cattle, and Drung cattle are precisely the same thing or they are considered different breeds of *Bos frontalis* species. While these four terms are used interchangeably in some sentences, they are referred to as distinct entities in some other sentences.

A response to this comment will help clarify the confusion created by the following parts of the manuscript

Line 501-504: "Chinese gayal probably descended from wild aurochs like modern cattle breeds. In contrast, Indian mithun was inclined to gather with gaur, suggesting that Indian mithun might be a descendant of gaur."

Comment 2: "Genome comparison with closely related species". In this section (lines 177-194), a phylogenetic tree is constructed to understand the separation and diversification of the bovines. One major concern with this analysis is that the *Bos gaurus* species is not included despite its close evolutionary proximity to *Bos frontalis*. A high-quality genome of *Bos gaurus* is available on NCBI as ARS_UOA_Gaur_1 (GCA_014182915) for almost a year now. Would you please justify why you have not used this species?

Have you considered the possibility of introgression between species after speciation? Given the semi-wild nature of Drung cattle, the possibility of gene flow with other bovine species after speciation is very intriguing and may be worth exploring. For instance, I suggest you could use the ABBA-BABA test [Green et al., 2010] to look for introgression signatures along the genome. Alternatively, you could provide your reasoning and discuss why gene flow may be an unlikely scenario.

Walker et al. citation need to mention volume number (Volume II) and page numbers (1427 & 1431). However, this reference does not have any genetic data.

The paper cited by you (Ma et al. 2007, PMID: 17560527 DOI: 10.1016/S1673-8527(07)60045-9) actually states in the abstract: "The results also indicate that a great proportion of gayal bloodline was invaded by other species, and the protection of gayal is facing a formidable situation."

In addition, the same Ma et al., 2007 paper has the following text: "There are crossbreed descendants of gayal and yellow cattle from Dulong River Basin to Nu River Basin according to the investigation in the field. This was confirmed by the research results that some gayal were of *Bos taurus* or *Bos indicus* matriline origin. Surprisingly, 75% of the gayals did not originate from the gayal in matriline. Because of the limited sampling scale, the proportion might not be so accurate, but it indeed reflected a serious issue that the protection of gayal was facing a serious situation. Gayal population scale increased rapidly in the past, but its bloodline was invaded by other species, a great proportion of gayal did not remain in the gayal matriline origin. The situation could be the result of the following two factors: 1) domestication at other places made it much

easier for the gayal to affiliate with yellow cattle or gaur, and the gayal lived semidomestically, so it was possible to give birth to lots of crossbreed descendants. 2) female crossbreed descendants were able to reproduce, but male descendants could not[15]."

Another paper cited by you (Prabhu et al., 2019) also suggests a complex scenario when they state: "Our phylogenetic analyses suggested the presence of two maternal lineages for mithun. The occurrence of two types of mithun, one which is a descendant of gaur, while the other which is a hybrid of mithun bull and cattle, has also been indicated by Dorji et al. [11]. Similarly, Baig et al. [1] have reported three different haplotypes for mithun. The recent studies [3, 5, 6, 26] based on whole genome sequencing and SNP genotyping have supported all the three proposed hypotheses for the origin of mithun. Therefore, the phylogenetic status of mithun remains contentious. Perhaps, the evolutionary history of mithun is more complex as reported in the case of many other livestock species. A comprehensive phylogenetic analysis with more number of mithun, gaur and cattle samples particularly from India and China would improve the current understanding of the evolutionary history of mithun."

Hence, the possibility of introgression from other closely related species (such as *Bos gaurus*) is a genuine concern and is also of great importance to the Drung Cattle conservation efforts. The recent paper by Chen et al. (2020) (<https://www.pnas.org/content/117/45/28150>) has not been cited. However, they demonstrate the close evolutionary proximity of *Bos gaurus* and *Bos frontalis* using mtDNA sequences. Another important paper (Baig et al. 2013) also from the "State Key Laboratory of Genetic Resources and Evolution, Kunming Institute of Zoology, Chinese Academy of Sciences (CAS), 650223 Kunming, China" is also note cited: Mitochondrial DNA diversity and origin of *Bos frontalis* (<https://www.jstor.org/stable/24110670>). The authors should note that clade F1 and F2 within *Bos Frontalis* are identified in the Baig et al. 2013 paper.

Comment 2: See comments regarding main figures

Figure 1 has a lot of data shown in a single figure. My suggestion is that this figure can be replaced by a more straightforward figure which is easier to follow. The transcriptome, methylome, etc., circles can be organized separately by chromosome in the supplements.

Figure 3: The gene order blocks shown on scaffold 60 in Figure 3C are not explicitly identified in Figure 3A. Have you used the same color scheme to link figure 3A and 3C?

Figure 6: The pathway shown in Figure 6A is based on the KEGG database, or is it obtained from any particular source? Would you please provide a reference for the connections and membrane localisations depicted in this figure?

The genes that are presumed missing in Figure 6B and Supplementary Table 19 could be missing from the genome assembly. However, it is still possible that they are intact in the genome. Have you searched the sequencing read datasets for these "missing genes"? Do you find the pseudogene sequences? Would you please provide more details?

Comment 3: "Genome-wide DNA methylation patterns". This section (lines 382-423) seems to have very little connectivity to the text before and after this section. Can you please improve this text? For instance, a tentative link between DNA methylation and changes in the heart muscle is discussed later (in lines 616-620). Could you rewrite this section to highlight this link?

Alternatively, you could identify differentially methylated regions between tissues and link them with changes in gene expression. The role of DNA methylation in gene expression is widely acknowledged.

Comment 4:

You have mentioned, "Like zebu and yak, Drung cattle lost both MYH4 (clade IV) and MYH6 (clade V) during gene family evolution. As a pseudogene, MYH16 gene was missing in the other five selected species except buffalo. In addition, the clade XVI with six gene members were only present in Drung cattle."

Changes in MYH gene content in Drung cattle is an important result that could explain the difference in meat quality of Gayal and other cattle. Hence, it needs to be evaluated in detail. Have you searched the sequencing read datasets for these "missing genes"? Do you find the pseudogene sequences? Are these pseudogene sequences expressed in RNAseq of any of the tissues? Would you please provide more details?

Comment 5:

Line 754-756: "PAML was used to identify positively selected genes from different species. The maximum-likelihood method was used to estimate dN, dS and dN/dS."

More details regarding which tests were done using PAML are missing. Did you perform branch tests? Supplementary Data 6 provides P-values. Are these p-values the results of Likelihood ratio tests? Are you able to give the likelihood values of the tests done?

Minor comments:

1) line 3: "..specie that mainly inhabits the hill-forests of "Grand Canyon of the East", Yunnan.."

2) line 51: "..has grown from 77 individuals at risk of extinction in 1,986 to nearly 3,000 heads now.."

3) line 58: "In term of biological classification, Drung cattle belongs to species *Bos frontalis*.."

4)line 70: "..altitude of 1,170-4,964 meters, which gives them unique physiological features that.."

5)line 71-73: "Studies have showed that the level of hemoglobin of Drung cattle, as well as the number of red blood cells and white blood cells, were equivalent to those of yaks living on the Tibetan Plateau.". No citations are provided here. Which studies are you referring to in this sentence?

6)line 149-156: "In comparison with taurine cattle genome, Drung cattle genome overrepresented ~40% LINE-L1 repeats, and had ~87% more LINE-RTE(*BovB*) repeats (702,019 in *Bos frontalis* and 376,067 in *Bos taurus*)#; instead, both LINE-L2 and LINE-CR1 decreased by approximately 1.5-fold, and SINE-*BovA* repeats had a >88% reduction (976,749 in *Bos frontalis* and 1,839,497 in *Bos taurus*) than that in taurine cattle#, indicating that as the ruminant- or cattle-specific repeats, SINE-*BovA* expanded primarily in the *Bos taurus* genome, whereas LINE-RTE(*BovB*) expanded specifically in the Drung cattle."

Consider splitting and rewriting the above into shorter sentences at the hash (#) annotations in the line above.

7)You note differences in repeat content of the genome assemblies and attribute this to biological differences. However, repeat regions are known to be difficult to assemble regions of the genome. Could these differences be a result of differences in how well these repeats are assembled in these genomes? Have you done any due diligence to rule out the possibility of assembly quality differences confounding the biology? For example, can you demonstrate few examples of species-specific repeats are supported by raw read datasets?

8)line 201: ""..gene families were contracted in gayal (Fig. 2a). Among the four species of genius *Bos*, gayal showed more events of gene family expansion with a 50-100 increase in..".

In the above sentence, do you mean genus *Bos*?

9)Line 232-234: "Among our gayal 20,181 annotated genes, we found 885 genes belonged to the OR supergene family, of which 22 OR genes originating from SD were exclusive to Drung cattle."

You refer to *Bos frontalis* as both gayal and Drung cattle in the above sentence. Would you please use consistent terms throughout the manuscript? Either use just Drung cattle or gayal.

Alternatively, you can write it as gayal/Drung cattle throughout the manuscript.

10)Line 578-580: "The strong resistance of cold and humidity as well as high stress tolerance to the environment are the remarkable features of adaptability in Drung cattle."

Change the above sentence to "The strong resistance to cold and humidity as well as high-stress tolerance to the environment are the remarkable features of adaptability in Drung cattle."

11)Line 695-696: "Clean reads were mapped to the *Bos frontalis* genome and *Btau_5.0.1* assembly using Bowtie and TopHat, respectively." Why are two different read mappers used for the two genomes? Have you used bowtie read mapper for the transcriptome of *Bos frontalis*? Would you please correct this sentence?

12)Line 699-700: "Repeat sequences were identified with de novo-based and homolog-based methods. The de novo gayal repeat library were build using Piler139 and RepeatModeler programmers."

Change the above sentence to "Repeat sequences were identified with de novo-based and homolog-based methods. The de novo gayal repeat library was built using Piler and RepeatModeler programs."

13)Line 748: "Calibration time was obtained from the TimeTree database.". How was the calibration time used to estimate split times in the phylogenetic trees constructed by PhyML? Are the split times from the time tree phylogeny?

14)Few other potential typos that have been corrected:

Line 7: environmental adaption have not been
Line 47: Drung cattle were included
Line 50: the Chinese
Line 52: that Drung cattle were domesticated
Line 68: *Bos frontalis* had relatively lower completeness
Line 69: Drung cattle inhabit the typical alpine valleys
Line 79: the gayal remains hitherto unclear
Line 135: autosomes
Line 303: we propose an evolutionarily conserved
Line 315: carries out
Line 328: had a strong association
Line 329: with certain tissues
Line 330: functional modules had corresponding unique tissues, respectively
Line 333: modules are fundamental
Line 334: in the heart relative to other
Line 371: hub genes were constructed
Line 373: involved in the circulatory system and muscle
Line 405: in the heart
Line 409: preferences for methylation were observed
Line 414: tends to use adenine (A) in non-CG contexts preferentially
Line 423: lowest in the lung and
Line 438: voltage-gated
Line 450: Drung cattle were positively selected
Line 455: the combined effects
Line 492: To date, the origin
Line 496: from the crossing of wild gaur
Line 497: . The research on [full stop missing before the beginning of a new sentence]
Line 518: at least one million years earlier
Line 549: So far, there is
Line 556: which frequently occurred during
Line 568: promotes the evolution of the Bovidae
Line 570: understanding of the karyotype reorganization
Line 579: high-stress tolerance to the environment, are the
Line 586: olfactory receptors (OR)
Line 600: odor perception in food-seeking
Line 603: crucial to understand the evolutionary responses of organisms better
Line 604: Frisch et al.
Line 607: cellular responses to higher-order phenotypes
Line 616: expressed genes of Drung cattle
Line 623: torrid, humid, and rainy climates.
Line 635: histology gives us
Line 636: biological processes related to
Line 641: the expansion of the MYH gene family
Line 645: significant differences between affected and unaffected animals
Line 656: insights into the molecular mechanism

Supplementary Material

Line 190: TRF version not mentioned
Line 209: Splice junction
Line 233: orthology/paralogy
Line 385: magenta module

In particular, please note that the following revisions would be necessary for us to
contact our referees again:
(1) Please analyze the potential for introgression between species, as suggested by
Referee #3, and include *Bos gaurus* in comparisons. We also would ask that you
include the centromeric fusion analysis recommended by Referee #1, and comment on
potential MYH pseudogenes or the validity of species-specific repeat regions (per
Referee #3).

**AU:** We sincerely appreciate the editor and referees for all supports, as well as the
main concerns about our works.

Regarding the analysis of introgression between species after speciation, as suggested
by Referee #3, Wu *et al.* carried out the genetic introgression in the *Bos* species
complex, including taurine cattle, zebu, gayal (*Bos frontalis*), gaur (*Bos gaurus*),
banteng, yak, wisent and bison, by examining the genomic regions of potential
introgression and detecting the direction of introgression (Wu *et al.*, *Nat Ecol Evol*
2018)¹. The result showed that the genetic introgression was determined from
domestic cattle to wisent and yak, from zebu to gayal and banteng, and from yak to
Tibetan cattle in Fig. 1 of the published paper. In view of the fact that the relevant
research results have been reported, we have cited the reference in the revised
manuscript. Please see lines 693-696.

Fig. 1: Phylogenetic tree and genetic introgression of species in the *Bos* genus (quoted from Wu DD
*et al. Nat Ecol Evol*, 2018) ¹.

**References:**

1 Wu, D. D. *et al.* Pervasive introgression facilitated domestication and adaptation in the *Bos*
species complex. *Nat Ecol Evol* **2**, 1139-1144, doi:10.1038/s41559-018-0562-y (2018).

As recommended by Referee #1, we have added more details of the centromeric
fusion analysis. First, we stated that Fragscaffold60 is indeed part of BFR1 that
contains the centromeric region. Secondly, we described the sequence of
Fragscaffold60 in detail. Fragscaffold60 has a total length of 34.4 Mb and contains
352 gaps. The size of all gaps is 617 Kb, comprising 1.8% of the total length. Thirdly,
we added Supplementary Data 17 that showed the order and orientation of 16 genes
on the Fragscaffold60 comparing to their homologous ones on the genome of bovine.
Finally, we focus on the Rob (2;28) chromosomal fusion in gayal, involving BTA2,
BTA28 and Fragscaffold60. By analyzing satellite DNA sequences and HOR-like
arrays in the centromere regions, we hypothesized the possible mechanism of the
formation of gayal chromosome 1. Regarding the specific questions raised by Referee
#1, we answered them one by one.

Referee #3 commented on potential MYH pseudogenes and the relevant expression of
in RNA-seq of any of the tissues. In the analysis of MYH gene family, we did search
for the "missing gene" sequences from the sequencing read datasets. A detailed
description of each gene member in the MYH gene family of Bovinae species is
showed in the Supplementary Data 22. In addition, the expression of pseudogene
sequences was not detected in the RNA-seq of any of the tissues, while other MYH
genes had their respective expressions in tissues.

(2) If feasible, we strongly encourage you to include additional replicates for
transcriptomic and methylome analyses, as suggested by Referee #2. At an absolute
minimum, we would ask that you address the sample size as a limitation in the

Discussion and qualify any statistical comparisons.

**AU:** Thank you for this concern. It should be necessary to point out that the
transcriptomic and methylome analyses were based on only one biological replicate,
which may lead to some deviations in our results. So, we do need to be cautious when
drawing conclusions. However, Drung cattle is an endangered species of cattle in
China with the scarcity in number and it has been preserved from the threat of
extinction, making it more difficult to obtain more biological replicate especially the
need for slaughter. In the Discussion section of the revised manuscript, we have added
the limitations in this regard, see lines 856-860. In the future, we would like to
validate our results following your advice if we have chance to gain more samples.

(3) Carefully proofread the manuscript for grammatical errors and clarity, as best
highlighted by Referees #2-3. In particular, it would be necessary to clarify whether
mithun, Drung cattle, and gayal are considered the same, or separate entities (as noted
by Referee #3). At the same time, we ask that you expand the Introduction and
Methods, as requested by the reviewers, to improve reproducibility of these analyses
and the context of the current study.

**AU:** Thank the reviewers for helpful suggestions. The revised manuscript has been
proofread by an English language editing agency (American Journal Experts, AJE).
According to the comments of the reviews, we have made major revisions to the
Introduction and Methods, and added some details in the Supplementary Notes. We
apologize for vague statements in the article and have clarified relevant issues. The
gayal (*Bos frontalis*), also known as mithun, is distributed in Northeast India,
Bangladesh, Myanmar and in Yunnan, China. It should be particularly pointed out that
Drung cattle is the Formal Name for Chinese gayal given by the Chinese agricultural
department, and is named after its unique distribution in the Drung River area of
Yunnan Province in China. Thus, Drung cattle is the same entity with Chinese gayal.

Reviewers' comments:

**Reviewer #1 (Remarks to the Author):**

In the paper entitled “A draft genome of Drung cattle (*Bos frontalis*) reveals clues to
its chromosomal fusion and environmental adaption” by Chen et al., the authors
assembled a better reference genome of Chinese specific Drung cattle, which can be a
good genetic resource of further genetic research and conservation of the Drung
cattle. Most importantly, this research revealed the formation mechanism of
Robertsonian chromosome in *Bos frontalis*. The overall description of the article is
rigorous and explicit. Below is a few issues and concerns that can make this
manuscript better and more applicable to be published on Communications Biology.

Major issues:

1. line 62: the reference of domesticated yak genome, refer to the newly published
version by Zhang et al. doi: 10.1093/molbev/msab134; Importantly, the authors
should note that *Bos mutus* is wild yak and *Bos grunniens* is domestic yak, you
misused it broadly in the paper. If needing to referring to the genome of wild yak, see
Liu et al., doi: 10.1038/s41597-020-0400-3.

**AU:** Thank you very much for pointing out the differences in reference genomes
between wild yak and domestic yak. We have already made corresponding corrections
according to the suggestion of the reviewer when quoting the literature.

2. line 75; Figure S3: Did the result conduct by authors? I didn't see any linked
method description. If not, add reference to original research.

**AU:** Yes. Figure S3 was obtained by routine hematoxylin and eosin staining (H & E)
on the samples we collected. And a method description has been added in the
Supplementary Notes, see lines 22-24.

3. Figure 1.2: The description is ambiguous; the blocks seem to be all the same.

**AU:** Figure 1.2 shows the synteny relationship aligning the scaffolds of our assembly

to *Bos taurus* genome. Each block represents a chromosome, and the length of the
block is proportional to the size of each chromosome.

4. lines 249-255: As the authors described, I suppose that Fragscaffold60 seemed to
be partial BFR1 that contains the centromeric region? The authors need to add
statement on this as ones may think that scaffold60 is BFR1.

**AU:** As your comment pointed out, Fragscaffold60 is indeed part of BFR1 that
contains the centromeric region. According to the suggestion of the reviewer, we have
added this statement. Please see lines 327.

5. Below are a few concerns on the result of chromosome fusion in Drung cattle.

1) Assembled sequences in centromeric region always contains gaps because its
property of highly repetitive. How about Fragscaffold60 in Drung_v1.2?

**AU:** Fragscaffold60 has a total length of 34.4 Mb and contains 352 gaps. The size of
all gags is 617 Kb, with the maximum length of 21,014 bp, the minimum length of 3
124 bp, and the average length of 1,823 bp.

2) There were only three copies of repetitive blocks. Except for Fragscaffold60, did
any other scaffolds contain similar blocks? Was the copy of CNEP-B box contracted
in Drung cattle and how's the affect to chromosome stability?

**AU:** It is well known that centromeres are mainly composed of satellite DNA. At
centromeric regions in human, satellite monomers are hierarchically organized into
larger repeating units, which are named "higher-order repeats" (HORs). HORs are
tandemly arranged into chromosome-specific satellite arrays with nucleotide
differences between repeat copies (Schueler et al. 2001; Sullivan and Sullivan
2020)^{2,3}. In the analysis of chromosomal fusion in gayal Rob (2;28), we found that the
centromeric regions of BTA2, BTA28 and Fragscaffold60 had similar characteristics,
and defined the HOR-like array as "block". Each block is composed of a multimeric
satellite array, in which the constituent monomers are distinguished (greater than 85%

sequence identity). In this context, three satellite blocks were detected in the
centromeres of BTA2, BTA28 and Fragscaffold60. As the reviewer commented, in
addition to Fragscaffold60, other scaffolds also contain similar block of HOR-like
structure by scanning Bovidae Satellite I DNA sequences. We have been carrying out
the comprehensive characterization and analysis of repetitive centromeres in gayal
and other bovine species. In this article, we are concerned with Rob (2;28)
chromosomal fusion in gayal referred to BTA2, BTA28 and Fragscaffold60.
The CENP-B box motif has been demonstrated in a wide range of species. Ana
Escudeiro et al. recently identified CENP-B box-like motif in three bovid tribes⁴. We
also found the presence of the CENP-B box-like in the satellite monomer sequence of
BTA2, BTA28 and Fragscaffold60. In the study of marmoset alpha satellite, the
CENP-B box sequence was at approximately a frequency of 1 in 3 repeat units⁵.
Similarly, sequence comparison revealed that the CENP-B boxes were located on two
positions in the satellite monomer of BTA2, BTA28 as well as Fragscaffold60, and the
array of CENP-B boxes along satellite DNA was highly associated with higher-order
repeat structures. The centromere is the essential chromosomal structure necessary for
accurate chromosome segregation during cell division⁶⁻⁸. CENP-B boxes are the
binding site for CENP-B (Centromere Protein B), the only sequence-specific
centromeric DNA-binding protein identified so far. As CENP-B binds to the CENP-B
box in a sequence-dependent manner, the interaction may organize arrays of
centromere satellite DNA into a higher-order structure, which then directs centromere
formation and the assembly of specific centromere structures in mammalian
chromosomes^{9,10}. Beside its role in centromere function, the binding of CENP-B to
CENP-B boxes in the satellite repeats can be able to preserve centromere integrity by
regulating DNA synthesis and replication fork stall¹¹. Consequently, the presence of
CENP-B boxes plays a pivotal role in contributing to centromere function in
chromosome segregation and maintaining genome stability.

**References:**

2 Schueler, M. G., Higgins, A. W., Rudd, M. K., Gustashaw, K. & Willard, H. F. Genomic and

genetic definition of a functional human centromere. *Science* **294**, 109-115, doi:DOI
10.1126/science.1065042 (2001).

Sullivan, L. L. & Sullivan, B. A. Genomic and functional variation of human centromeres.
*Exp Cell Res* **389**, 111896, doi:10.1016/j.yexcr.2020.111896 (2020).

Escudeiro, A., Adegas, F., Robinson, T. J., Heslop-Harrison, J. S. & Chaves, R. Conservation,
Divergence, and Functions of Centromeric Satellite DNA Families in the Bovidae. *Genome*
*Biology and Evolution* **11**, 1152-1165, doi:10.1093/gbe/evz061 (2019).

Suntronpong, A. *et al.* CENP-B box, a nucleotide motif involved in centromere formation,
occurs in a New World monkey. *Biol Letters* **12**, doi:ARTN 20150817
10.1098/rsbl.2015.0817 (2016).

Balzano, E. & Giunta, S. Centromeres under Pressure: Evolutionary Innovation in Conflict
with Conserved Function. *Genes-Basel* **11** (2020).

Thakur, J., Packiaraj, J. & Henikoff, S. Sequence, Chromatin and Evolution of Satellite DNA.
*Int J Mol Sci* **22**, doi:ARTN 4309
10.3390/ijms22094309 (2021).

Suzuki, Y., Myers, E. W. & Morishita, S. Rapid and ongoing evolution of repetitive sequence
structures in human centromeres. *Sci Adv* **6** (2020).

Morozov, V. M., Giovinazzi, S. & Ishov, A. M. CENP-B protects centromere chromatin
integrity by facilitating histone deposition via the H3.3-specific chaperone Daxx. *Epigenet*
*Chromatin* **10** (2017).

Gamba, R. & Fachinetti, D. From evolution to function: Two sides of the same CENP-B
coin? *Experimental Cell Research* **390** (2020).

Barra, V. & Fachinetti, D. The dark side of centromeres: types, causes and consequences of
structural abnormalities implicating centromeric DNA. *Nat Commun* **9** (2018).

12 Ma, G. L. *et al.* Phylogenetic relationships and status quo of colonies for gayal based on
analysis of Cytochrome b gene partial sequences. *Journal of Genetics and Genomics* **34**, 413-419,
doi:10.1016/s1673-8527(07)60045-9 (2007).

13 Baig, M. *et al.* Mitochondrial DNA diversity and origin of *Bos frontalis*. *Curr Sci India* **104**,
115-120 (2013).

14 McIntosh, B. B. & Ostap, E. M. Myosin-I molecular motors at a glance. *J Cell Sci* **129**, 2689-
2695, doi:10.1242/jcs.186403 (2016).

3) How was the order and orientation of genes of Drung_v1.2 on the scaffold60
comparing to their homologous ones on the genome of bovine? This is worth
verifying to further support the centromeric fusion model.

**AU:** Thanks for your suggestion. The scaffold60 contains 16 genes with an average
length of 23,613bp, ranging from 435bp to 11,1071bp, comparing to their

homologous ones on the genome of bovine. We added **Supplementary Data 17** that
described the order and orientation of the 16 genes in detail. The gene alignment
legend was illustrated on the right side of Figure 3a.

6. Below are a few consents on the result of the biological process.

1) line 434: expansion is a relative conception. I suggest authors to list the related
gene families containing expanded genes and compare the number of genes clustering
in which across all mentioned mammals.

**AU:** Thanks for your suggestion. In the circulation and contraction-related biological
process, there are 22 expansion genes, which contains 13 MYH gene family members
(**Supplementary Data 21**). MYHs are the major contractile proteins that bind to actin
and hydrolyze adenosine triphosphate to provide force for diverse movements such as
cytokinesis, phagocytosis, and muscle contraction. In the biological pathway analysis
associated with adaption, the MYH gene family is likely to play the most critical role
in the adaptation of Drung cattle. Therefore, we compared the number and function of
the MYH gene family in six species of Bovinae (**Supplementary Data 22**), please see
lines 632-649.

2) line 479: genes in the clade XVI seems have massive genetic variations with genes
in other clades. Are they under positive selection in Drung cattle? If they share
nonsynonymous mutations comparing to others, they may take important role in
adaption in Drung cattle.

**AU:** In this study, we totally identified 1,102 positively selected genes (PSGs).
Detailed PSGs information were described in **Supplementary Data 14** and the clade
XVI with six gene members was not included. Therefore, genes in the clade XVI of
Drung cattle are not subject to positive selection, and the shared nonsynonymous
mutations cannot be obtained.

3) Are there similar researches on the expansion genes that the authors mentioned in

the paper, for example, the copy number affect the function.

**AU:** Thanks for your suggestion. Rutland *et al* revealed lineage-specific duplications
of MYH genes in the chicken genome and confirmed that MYH gene expansions
were largely avian specific, suggesting that at the point of divergence of the diapsids
(birds, lizards) and synapsids (mammals) a small MYH gene family existed¹².
Previous studies have found that *ROCK2*, *MYH7*, *MYH7B*, *MYH9* had copy number
variations associated with cardiac development or heart and liver disease¹³⁻¹⁶. In
particular, the distribution of *MYH3* copy numbers in Chinese cattle breeds was
detected and validated, and the results further demonstrated the association of the
copy number changes with its transcriptional expression and cattle growth traits^{17,18}.

**References:**

- 12 Rutland, C. S. *et al*. Knockdown of embryonic myosin heavy chain reveals an essential role
in the morphology and function of the developing heart. *Development* **138**, 3955-3966,
doi:10.1242/dev.059063 (2011).
- 13 Singer, E. S. *et al*. Characterization of clinically relevant copy-number variants from exomes
of patients with inherited heart disease and unexplained sudden cardiac death. *Genet Med* **23**, 86-
93 (2021).
- 14 Ma, Y. S. *et al*. Proteogenomic characterization and comprehensive integrative genomic
analysis of human colorectal cancer liver metastasis (vol 17, 139, 2018). *Mol Cancer* **18** (2019).
- 15 Molck, M. C. *et al*. Genomic imbalances in syndromic congenital heart disease. *J Pediat-*
*Brazil* **93**, 497-507 (2017).
- 16 Haraksingh, R. R. *et al*. Exome sequencing and genome-wide copy number variant mapping
reveal novel associations with sensorineural hereditary hearing loss. *BMC genomics* **15** (2014).
- 17 Zhang, L. Z. *et al*. Detection of copy number variations and their effects in Chinese bulls.
*BMC genomics* **15** (2014).
- 18 Xu, Y. *et al*. Associations of MYH3 gene copy number variations with transcriptional
expression and growth traits in Chinese cattle. *Gene* **535**, 106-111 (2014).

7. lines 494-504: For the domestication and phylogeny of *Bos* genus species, Zhang et
al., doi: 10.1111/age.12974, presented a comprehensive review, the authors may refer
to it.

**AU:** We sincerely appreciate your helpful suggestion. This reference has been cited in
line 668.

Minor issues:

1. line 51: year number is comma-delimited, while some numbers were not comma-
delimited in supp. note.

**AU: Revised.**

2. lines 148-156: need a space between LINE-RTE and (BovB). There were many
similar issues in the main text and supp. note, i.e. space between text and left
parenthesis.

**AU: Revised.**

3. line 1325 Figure 2b, it is an upset plot, not venn plot.

**AU: Revised.** We have changed the “Venn diagram” to “Upset plot”. Please see line
260.

4. line 1386: add the label to y-axis in Figure 5e.

**AU: Revised.**

5. In Method and Supplementary Note section, the authors should add the version of
all software and tools.

**AU: Revised.**

6. line 185 and line 217: use consistent expression of e-value cutoff.

**AU: Revised.**

7. lines 228-229: Add the resources or references to these genome sequences.

**AU: Revised.**

**Reviewer #2 (Remarks to the Author):**

The authors performed the de novo genome sequencing of a Drung bull by combining
Hiseq&Miseq, PacBio long-read sequencing, and 10x Genomics platforms to generate
an improved *Bos frontalis* genome assembly. To evaluate the quality of a new
assembly, several methods have been used, which include contig and scaffold size
measurements (contig N50, scaffold N50), gene set completeness, synteny
comparison, and the Benchmarking Universal Single-Copy Orthologs (BUSCO).
Next, they performed several analyses to better understand Drung cattle origin and its
taxonomic status, gene duplication, and biological process of local adaptation, and the
mechanism of chromosomal fusion at the genome-scale. Finally, a combined analysis
of transcriptomics and DNA methylation of Drung cattle was carried out to identify
functional genes and understand the molecular biological mechanisms associated with
animal adaptation in the challenging ecology of alpine valleys.

Overall, they used high-level technology to sequence the genome of gayal and they
reported improvement in genome assembly which is good. However, many
comparisons are made using the methylation and transcriptome data but most of them
lack a proper assessment of statistical significance as only one biological replicate has
been used. This is a strong limitation to draw reliable conclusions from the tissue
comparison. Also, at several occurrences the methods should be clarified, better
documented, and cleared of potential biases. In the introduction section, there is a bit
of historical information and would benefit from focusing more on problems of
existing genome assembly for gayal.

My review is divided into two parts: (A) Major general issues with detailed
comments, and (B) Minor issues.

**A) Major issues**

1) Introduction needs to be revised completely. Much effort has been put into

describing the habitat and history of the Drung cattle while focusing less on problems
of existing genome assembly. It is necessary to explain factors affecting the accuracy
and continuity of the genome assembly and how the current study is supposed to
address those problems.

**AU:** Thanks for your comment and suggestion. We have carefully revised the
introduction section. Firstly, we remove some unnecessary descriptive statements
about Drung cattle. The problem with the existing Drung cattle assembly genome is
the relatively lower completeness compared to other bovine species, especially the
smaller sizes in scaffold N50 and contig N50. Meanwhile, the genetic basis of its
chromosome fusion and adaptability remains largely unknown. Secondly, we give a
supplementary description to this part, see lines 95-101. Thirdly, we describe the
factors that influence the accuracy and continuity of the genome assembly as the
reviewer's recommendation, including the length of reads, size of the library, accuracy
of reads, uneven sequencing depth and complexity of the genome, see line 102-111.
Sequencing technology, assembly approaches, quality control, and bioinformatics
strategies are continually being developed to overcome these problems. The
emergence of innovative, disruptive technologies such as long-read sequencing,
linked reads, Hi-C and optical mapping have greatly driven the improvement of a
high-quality genome assembly. In the revised introduction, we briefly describe the
sequencing technologies used and give examples of similar studies. PacBio long read
sequencing gives much better resolution and contiguity as it can span satellite repeats
and segmental duplications as well as provide unambiguous links between
nonrepetitive regions. 10x Genomics linked-read sequencing has the efficacy in
resolving long and highly similar repetitive regions by filling gaps and guiding the
order and orientation in fragmented scaffolds and contigs. Here, we generate an
improved *Bos frontalis* genome assembly by adopting a hybrid *de novo* assembly
approach using a combination of short reads from Illumina HiSeq & MiSeq platform,
long reads from PacBio platform, and linked-reads from 10x Genomics platform.

2) Method section writing should be improved. The writing should be improved to
better explain the methods. Several steps of the analysis are not clear or insufficiently
documented. It is not clear how sequencing, library preparation, and quality control
have been done for genome and transcriptome sequencing and methylation calling. It
is necessary to know the coverage and the number of reads for different tissues.
Several statistical tests have been performed without providing enough details about
statistical tests, significance levels (p-value), etc. It is not clear which database has
been used for downloading genes and protein sequences for each species. In the gene
ontology analysis, it is not clear what background set has been used. The version of
all used software has not been documented.

**AU:** Thanks for the concerns about the Method section. Not all detailed materials,
methods, and software are presented due to the word limit required by the journal. We
give a brief description of the materials and methods in the text of the manuscript, and
put the relevant details in the Supplementary notes. I would like to clarify your
specific question as follows.

**Q1:** The detailed methods of library preparation, sequencing, and quality control for
genome and transcriptome sequencing and methylation calling are described in the
**Supplementary Notes**, see lines 26-78.

**Q2:** The coverage and the number of reads for different tissues are shown in the
**Supplementary Data 9**.

**Q3:** The statistical tests and significance levels are provided in Method section (line
1,018, line 1,029) and Supplementary notes (line 376).

**Q4:** The genes and protein-coding sequences of the ten mammalian species were
downloaded from NCBI database, and the annotation of our *Bos frontalis* assembly
genome were used to define gene families, see lines 745-748 in Method section.

**Q5:** In the gene ontology analysis, the protein coding genes of cattle (*Bos taurus*) was
used as the background set.

**Q6:** We have added the version information of all used software in the revised
manuscript.

3) Incompatibility between results of the phylogenetic tree and previous speciation
studies. The authors estimated the divergence time based on single-copy gene families
and reported the divergence time of *Bos taurus* from *Bos indicus* at 1.8 Mya (1.3-2.5).
They also reported that *Bos frontalis* evolved ~1.20 Mya earlier than domestic cattle.
However, these estimates are a bit far away from previous estimates. For example, a
previous study (Wang et al 2017) estimated the divergence time of gayal from cattle
and zebu at approximately 5.1 million years ago. Also, previous studies using
mitochondrial genome, microsatellite, etc reported divergent times of *Bos taurus* from
*Bos indicus* less than 1 million years. The authors should discuss how confident are
their results? How much is the resolution of estimates? Compare current results with
previous studies.

**AU:** We appreciate you for this comment. We reported that the divergence time of *Bos*
*taurus* from *Bos indicus* is at 1.8 Mya (1.3-2.5) based on single-copy gene families,
which is consistent with the result by the analyses of the complete *Bos taurus* and *Bos*
*indicus* mitochondrial genome sequences (Hiendleder *et al.*, 2008). They demonstrated
that the estimated divergence times of the two cattle lineages separated 1.7-2.0 million
398 years ago¹⁹. Both our research and Hiendleder *et al.*, respectively, used more than two
reference times for calibrating the molecular clock. In our study, the divergence time of
each node was calibrated against the fossil calibration times for the *Homo sapiens*-*Bos*
*taurus* divergence (95.3–113 Mya) and *Bos taurus*-*Ovis aries* (18.3–28.5 Mya). Many
previous studies have assessed the relevant divergence time. For example, Pramod *et*
*al* (2019). sequenced and analyzed the mitochondrial DNA (mtDNA) of seven Indian
cattle breeds, showing that the *Bos indicus* and *Bos taurus* cattle lineages diverged 0.92
million years ago²⁰. They estimated the divergence time for other species by fixing only
one divergence time between *Capra hircus* and *Ovis aries*. The presumed explanation
for this time difference may be the different reference points used, or it may be two
independent primary domestication events from genetically discrete aurochs groups.
This is also discussed in detail in the article of Pramod *et al* ²⁰. Wang *et al* (2017)

estimated the divergence times among bovine species by fixed calibration time of
buffalo and cattle. Although their study showed that the divergence time of gayal from
cattle and zebu at 5.1 (3.9-5.8) million years ago, it could be noted that the divergence
time of cattle from zebu are estimated to be 3.5 (2.6-4.2) million years ago, which is
much higher than estimates from other studies²¹. Nevertheless, it can be deduced that
gayal evolved 1.3-1.6 Mya earlier than cattle and zebu, in close proximity to our
estimate of 1.2 Mya.

**References:**

19 Hiendleder, S., Lewalski, H. & Janke, A. Complete mitochondrial genomes of *Bos taurus* and
*Bos indicus* provide new insights into intra-species variation, taxonomy and domestication.
*Cytogenet Genome Res* **120**, 150-156, doi:10.1159/000118756 (2008).

20 Pramod, R. K. *et al.* Complete mitogenome reveals genetic divergence and phylogenetic
relationships among Indian cattle (*Bos indicus*) breeds. *Anim Biotechnol* **30**, 219-232,
doi:10.1080/10495398.2018.1476376 (2019).

21 Wang, M. S. *et al.* Draft genome of the gayal, *Bos frontalis*. *GigaScience*,
doi:10.1093/gigascience/gix094 (2017).

4) Lack of significance for transcriptomic and methylation study. Only one biological
replicate is included for all tissues analyzed in the study. This limits the proper
conclusion from tissue comparison and the author needs to take caution in this regard.
For example, in abstract lines 27 the authors have reported the lowest methylation for
the heart. How did you conclude this with only one biological replicate per tissue? How
did you calculate the P-value? This limitation ought to be indicated somewhere in the
manuscript though. I think it would be great if the author could validate their results.

**AU:** Thanks for this concern. First, transcriptome data from different tissues is
necessary for the annotation of the Drung genome. Next, in order to understand the
molecular mechanisms of tissue-specific gene expression and regulation, we performed
comparative transcriptome analysis and weighted gene co-expression network analysis,
and identified the heart-specific gene co-expression module as well as individual hub
genes (e.g., *RYR2*, *TNNI3*, and *ACTC1*) underlying the important biological processes
such as circulatory system process and voltage-gated channel activity. In addition, we
observed that the overall methylation level of the heart was different from that of other

tissues. Taken together, we therefore focused on the identified evolutionary genes to
gain insight into the molecular genetic basis of cardiac adaptive evolution. In fact, as
the reviewer's concern, it should be necessary to point out that this part of our results
was based on only one biological replicate and cannot be calculated with p value, which
may lead to some deviations in the results. So, we do need to be cautious when drawing
conclusions. Considering that Drung cattle is an endangered species of cattle in China
with the scarcity in number, it has been preserved from the threat of extinction, making
it more difficult to obtain more biological replicate especially the need for slaughter. In
the Discussion section of the revised manuscript, we have added the limitations in this
regard, see line 856-860. In the future, we would like to validate our results following
your advice if we have chance to gain more samples.

**B) Minor issues**

P.3, 1.12- “Bosin” => Bison

**AU:** Revised. Please see lines 92.

P.3, 1.13- “MYa” all abbreviation should be explained at first => Million years ago

(MYa)

**AU:** Revised.

P.3, 1.14- “suggesting ...”, => “confirming ..”

**AU:** Revised.

P.3, 1.16- “We found a consensus conserved CENP-B...”, I would suggest saying “we

confirmed ... “, as a previous study had been reported it in the Bovidae. (Escudeiro,

468 A., Adegá, F., Robinson, T.J., Heslop-Harrison, J.S., and Chaves, R. (2019)

Conservation, divergence, and Functions of Centromeric Satellite DNA Families in

the Bovidae. *Genome Biology and Evolution*. 11 (4), 1152–1165)

AU: Revised.

p.5, 1.51- “1,986” => 1986

AU: In the manuscript, we have unified the number format by kilobit separators.

p.5, 1.66-68- “In comparison with the above-mentioned bovine species, the two
published genome assemblies of *Bos frontalis* had a relatively lower completeness.” It
is better to explain in more detail.

AU: Revised. Please see lines 168-169. And the two published genome assemblies of
*Bos frontalis* information were shown in Table 1.

p.6, 1.74-78- “Compared to the other native cattle in Yunnan, the muscle fibers of
Drung cattle are smaller in diameter, higher in density and larger in
number capillaries”. I didn’t find anything in the method section about
these comparisons. Did the author did this by themselves or this is from another study,
so please provide the reference of the study or document it in the method section.

AU: The result was conducted by authors. And a method description has been added
to the Supplementary Notes. Please see lines 22-24.

P.6, 1.82- “2n=58” => 2n=58 karyotype

AU: Revised. Please see lines 89.

p.6, 1.95- “3-year-old male Drung cattle” => 3-year-old Drung bull

AU: Revised.

p.7, 1.104- “10XGenomics” => 10x Genomics

AU: Revised.

p.8, 1.133- remove the link

**AU:** Revised.

p.9, 1.172-173- How robust is your gene set considering only one biological replicate
you have?! what is the tissue specificity index? Please explain how was calculated?

**AU:** In this study, a sum of 18,176 (90.06%) annotated protein-coding genes were
expressed in at least one of the fourteen tissues examined by transcriptome
sequencing (RNA-seq), supporting the high accuracy of gene annotation. Considering
the limitations of the number of samples, the gene set provides a reference for
subsequent analysis. The tissue specificity index (τ) in an index of gene classification,
0 for housekeeping genes and 1 for tissue-specific genes. The τ value was calculated
as reference Yanai et al.

p.10, 1.205-208- “A further functional characterization ... and regulation of actin
cytoskeleton (Cell motility).” Please provide P-value for enrichment results.

**AU:** We have presented more detailed enrichment results including P-value, Q-value
in Supplemental Data 15.

p.11, 1.223-228- “The enriched GO categories (Supplementary Fig. 7) ... genes were
significantly enriched (268 of 297, 90.24%).” Please report all related P-values!

**AU:** We have presented more detailed enrichment results including P-value, Q-value
in Supplemental Data 16.

p.16, 1.377-378- “had high degrees of connection, and participated in more biological
processes than other hub gens” please report significant tests and P-values?

**AU:** Thank for your comments. Herein we cannot provide the significant tests and P-
values, and we revised the sentence.

p.18, 1.436- “DHPR family”. Stand for what? Please provide full names for all gene
abbreviations throughout the manuscript.

**AU:** DHPR stand for dihydropyridine receptor, a member of the family of voltage-
operated Ca²⁺ channels, also known as Cav1.1. We have already provided full names
for all gene abbreviations throughout the manuscript.

p.22, 1.563-566- please reference a previous study that discovered the CENP box in
cattle before your study and discusses it. “Escudeiro, A., Adegas, F., Robinson, T.J.,
Heslop-Harrison, J.S., and Chaves, R. (2019) Conservation, divergence, and
Functions of Centromeric Satellite DNA Families in the Bovidae. Genome Biology
and Evolution. 11 (4), 1152–1165”

**AU:** Revised, please see lines 737-755.

p.23, 1.584-588- “Our functional analysis of SDs in Drung cattle revealed the most
significant enrichment in genes were specific to Drung cattle.” Please
provide p-value and explain which kind of enrichment you did (BP, CP, MP)? explain
what is the background gene?

**AU:** We added to the P-value to the olfactory transduction, see in line 772. We used
the KEGG database to enrich analysis of the SD genes in Drung cattle. And the
background genes were all genes with pathway annotation (14,967). Meanwhile, we
have provided more detailed results information about SDs in **Supplementary Data**
**16.**

p.24, 1.610-612- Among the identified tissue-specific modules, the heart-specific
module presented a significant co-expressed gene clustering,” provide statistical test
and p-value within a parenthesis.

**AU:** We used the correlation coefficient greater than 0.90 and P-values of the Fisher's
exact test less than 0.01 to identify tissue-specific modules, and provided statistical
test and p-value within a parenthesis in lines 1029.

p.24, 1.616-618- “The whole genome DNA methylation profiling showed the lowest

level in the heart with more non-CG DNA methylation compared to other tissues.”
please provide statistical test and p-value within a parenthesis. How significant and
precise is this considering you used only one biological rep.

**AU:** Because there was only one biological repetition in this study, statistical test
cannot be carried out to calculate the p value. The mC percent was calculated: mC
563 percent (%) = mC/Covered C*100. We compared the mC percent among the four
tissues, and the lowest level in the heart (2.75%) and the highest level in the muscles
(3.29%). Details of the conclusions for the heart tissue with the lowest methylation
were shown in **Supplementary Data 20**.

p.26, 1.660- “Genome sequencing and assembly” please provide more information
about the way library preparation, sequencing, and quality control have been done!

**AU:** We have already presented more detailed information about the library
preparation, sequencing, and quality control for genome sequencing and assembly in
**Supplementary Notes**. Please see lines 25-78 and lines 80-141.

p.27, 1.689- “Transcriptome sequencing” please provide more information about the
way library preparation, sequencing, and quality control has been done!

**AU:** We have already presented more detailed information about the library
preparation, sequencing, and quality control for transcriptome sequencing in
**Supplementary Notes**. Please see lines 66-78 and lines 143-156.

p.27, 1.694-696- “Clean reads were mapped to the Bos frontalis genome and
Btau_5.0.1 assembly using Bowtie136 and TopHat137, respectively.” Why two
mappers have been used? Did you use the default parameter for mapping with each?

**AU:** In this study, the Bowtie software was used to build-index for genomes instead
of reads mapping. The TopHat software was used to aligns RNA-Seq reads to
genomes with the default parameter.

p.27, 1.712-714- “We then aligned protein sequences of *Bos taurus*, *Ceratotherium*
*simum simum*, *Bos mutus*, *Bison bison* and *Bubalus bubalis* to *Bos frontalis* by using
TBLASTN145 for homology-based gene prediction.” Please provide a reference to
the database for each species protein sequence you downloaded.

**AU: Revised.** We downloaded database for each species protein sequence from NCBI
(<https://www.ncbi.nlm.nih.gov/>).

p.27, 1714-716- “Annotation of the predicted genes was performed by blasting the
sequences against InterPro146, Gene Ontology147, KEGG148, Swissprot and
TREMBL149 database.” Please rewrite the sentences. To me, it seems you blast
sequence against all databases!! while Gene ontology analysis uses gene lists not
sequence as input!

**AU: Revised.** We have changed “Annotation of the predicted genes was performed by
blasting the sequences against InterPro, Gene Ontology, KEGG, Swissprot and
TREMBL database.” to “Annotation of the predicted genes was performed by
combining InterPro, Gene Ontology, KEGG, Swissprot with TREMBL database.”

p.28, 1.743-744- “The protein-coding genes of the ten mammalian species
downloaded from NCBI”. Did you use RefSeq or Ensembl? What is the release date?
etc Please give detailed information.

**AU: We used RefSeq in this study.** And we provided the protein-coding genes of the
ten mammalian species with detailed information. “We selected six species in the
Bovinae subfamily (*Bos frontalis* (Drung_v1.2), *Bos Taurus* (Btau_5.0.1), *Bos indicus*
(*Bos_indicus_1.0*), *Bos grunniens* (BosGru_v2.0), *Bubalus bubalis* (Bubbub1.0) and
*Bison bison* (Bison_UMD1.0)) and other five mammals (*Ovis aries*
(GCA_011170295.1), *Capra hircus* (GCA_015443085.1), *Pantholops hodgsonii*
(GCA_000400835.1), *Ceratotherium simum simum* (GCA_000283155.1) and *Homo*
*sapiens* (GCA_001524155.4)) as the outgroup for phylogenetic analysis of the *Bos*
*frontalis*.”. Please see lines 696-698.

p.29, 1.750-752- “The protein-coding genes set of the eleven mammalian species were
used to identify gene families that descended from a single ancestral gene in the last
common ancestor of the given species.”. the same as my previous comment. Please
reference all databases used.

**AU:** Revised.

p.29, 1.756- “dN, dS and dN/dS.” => please add explanation what is dN, etc?

**AU:** We have revised this section. Please see lines 1006-1019. Additionally, the dN
and dS mean the rate of non-synonymous substitution, the rate of synonymous
substitution, respectively. The dN/dS means the ratio of non-synonymous
substitutions to the rate of synonymous substitutions.

p.29, 1.758- “FPKM” => please add full name

**AU:** Revised.

p.29, 1.769-770- “WGBS libraries were constructed according to the manufacturer’s
protocol.” please provide more information about extraction, sequencing, methylation
calling software used and parameters.

**AU:** We have already presented more detailed information about the DNA extraction,
sequencing, methylation calling software used and parameters for WGBS analyse in
Supplementary Notes. Please see lines 27-37 and lines 420-441.

p.29, 1.772- “The methylation level” What is the methylation level? How has been
calculated?

**AU:** The methylated cytosines (mCs) covered by at least five reads were used to
calculate the methylation level. The methylation level was determined by dividing the
number of reads covering each mC by the total reads covering that cytosine (the
number of mC reads and the number of nonmethylation reads). Please see lines 442-

446 in Supplementary Notes.

p.29, 1.774- “mCG, mCHG, and mCHH” stand for what? Explain in the main text.

**AU:** In mammals and plants, DNA methylation occurs in mCG dinucleotide
sequence context, as well as in mCHG and mCHH sequence contexts, where H stands
for all nucleotides except guanine (Lindroth et al. Science 2001). And we explained
the three DNA methylation contexts (mCG, mCHG and mCHH) in the main text,
please see lines 740-742.

p.29, 1.776-777 How TSSs and TES were defined? please document which annotation
data you used?

**AU:** The transcription start point (TSS) is defined as the base on the DNA chain
corresponding to the first nucleotide of the mRNA chain during transcription, usually
a purine. The transcription start site (TES) refers to the base on the DNA chain
corresponding to the last nucleotide of the new RNA chain. In this study, the
annotation data were obtained from Drung cattle reference genome annotation file
(Drung_v1.2 version).

Figures- Generally, fonts are unreadable and need to be increased.

**AU:** Revised.

Fig2. To me, pie charts are not informative. What are you trying to add to the figures
by adding pie charts? I would suggest omitting them to simplify the figure.

**AU:** Thanks for your suggestion. This pie chart showed the ratio of gene family
expansion and contraction. We omitting pie charts to simplify the figure.

**Reviewer #3 (Remarks to the Author):**

Brief summary of the manuscript

The manuscript by Chen and Zhang et al. generates an impressive amount of genome,
transcriptome, and methylome data analyzed to identify unique biological aspects of
Drung cattle. A detailed summary of data generated and analysis conducted are
enumerated below.

Summary of data generated:

Chen and Zhang et al. generate short-read Illumina sequencing, long-read PacBio
sequencing, and linked-read 10XGenomics data. These data are assembled to a
relatively high quality (demonstrated by measures such as N50, BUSCO score, repeat
content) genome assembly of Drung cattle (*Bos frontalis*).

Genome annotation is done using ab initio prediction, homology-based
approaches, and primary transcriptome data (14 tissue samples) generated in this
study.

Genome-wide methylation levels of 4 tissues are generated. The methylation level
is estimated using Bismark software to obtain the landscape of genome-wide DNA
methylation.

Summary of analysis conducted:

The genome assembly is compared with genomes of other species to identify
changes (expansion/contraction) in gene content (CAFE analysis, upset plot, and GO
enrichment analysis).

The putative location of the Robertsonian fusion of BTA2 and BTA28 is
identified. Gene synteny analysis, repeat annotation, and pairwise genome alignments
analysis are used to try and reconstruct the sequence of events involved in this
karyotype change.

Gene co-expression networks are identified using the WGCNA approach.

Changes in gene content (prominent examples being the loss of MYH4 (clade IV)
and MYH6 (clade V)), the prevalence of positive selection, and changes in gene
expression patterns are identified in genes of the circulation and contraction-related
biological processes. These genes are proposed to be involved in the adaption of
Drung cattle.

Overall impression of the work

The manuscript is a significant contribution to further our understanding of the
genomics and basic biology of *Bos frontalis* (Mithun/Gayal/Drung cattle). The
datasets generated in this study will be of great value. The text is easy to follow and
well written (see minor comments for a list of suggested changes).

Authors should provide more details (see major comments) regarding the methods
used in the manuscript to support their novel results. Some additional analysis to
support the significant claims also seems to be required.

Few lacunae in the approaches used and missing references are listed in the comments
below.

Specific comments, with recommendations for addressing each comment

**Major comments:**

**Comment 1:**

The authors need to state if, according to them, the Chinese gayal, Indian Mithun,
Dulong cattle, and Drung cattle are precisely the same thing or they are considered
different breeds of *Bos frontalis* species. While these four terms are used
interchangeably in some sentences, they are referred to as distinct entities in some

other sentences. A response to this comment will help clarify the confusion created by
the following parts of the manuscript. Line 501-504: "Chinese gayal probably
descended from wild aurochs like modern cattle breeds. In contrast, Indian mithun
was inclined to gather with gaur, suggesting that Indian mithun might be a descendant
of gaur."

**AU:** The Chinese gayal, Dulong cattle, and Drung cattle are precisely the same breed.
We sincerely apologize for possible confusion in the description. "Dulong" cattle and
"Drung" cattle are the differences in English translation of the same Chinese word,
which have completely the same meaning. In order to avoid confusing readers, we use
"Drung cattle" uniformly in the revised manuscript. Drung cattle is the Formal Name
for Chinese gayal given by the Chinese agricultural department, named due to its
unique distribution in the Drung River area of Yunnan Province in China. Indian
Mithun refers to the distribution of gayal in India. Based on phylogenetic analysis, we
discuss the possible differences in the evolutionary origins of Chinese gayal and
Indian Mithun.

**Comment 2:**

"Genome comparison with closely related species". In this section (lines 177-194), a
phylogenetic tree is constructed to understand the separation and diversification of the
bovids. One major concern with this analysis is that the *Bos gaurus* species is not
included despite its close evolutionary proximity to *Bos frontalis*. A high-quality
genome of *Bos gaurus* is available on NCBI as ARS_UOA_Gaur_1
(GCA_014182915) for almost a year now. Would you please justify why you have not
used this species?

**AU:** Thanks for this concern. Although a high-quality genome of *Bos gaurus* is
available on NCBI as ARS_UOA_Gaur_1 (GCA_014182915), the CDS sequence of
the *Bos gaurus* genome is not available. So, we cannot construct a phylogenetic tree
containing *Bos gaurus* by extracting the single-copy orthologous genes.

Have you considered the possibility of introgression between species after speciation?
Given the semi-wild nature of Drung cattle, the possibility of gene flow with other
bovid species after speciation is very intriguing and may be worth exploring. For
instance, I suggest you could use the ABBA-BABA test [Green et al., 2010]) to look
for introgression signatures along the genome. Alternatively, you could provide your
reasoning and discuss why gene flow may be an unlikely scenario.

The paper cited by you (Ma et al.2007, PMID: 17560527 DOI: 10.1016/S1673-
8527(07)60045-9) actually states in the abstract: "The results also indicate that a great
proportion of gayal bloodline was invaded by other species, and the protection of
gayal is facing a formidable situation." In addition, the same Ma et al., 2007 paper has
the following text: "There are crossbreed descendants of gayal and yellow cattle from
Dulong River Basin to Nu River Basin according to the investigation in the field. This
was confirmed by the research results that some gayal were of *Bos taurus* or *Bos*
*indicus* matriline origin. Surprisingly, 75% of the gayals did not originate from the
gayal in matriline. Because of the limited sampling scale, the proportion might not be
so accurate, but it indeed reflected a serious issue that the protection of gayal was
facing a serious situation. Gayal population scale increased rapidly in the past, but its
bloodline was invaded by other species, a great proportion of gayal did not remain in
the gayal matriline origin. The situation could be the result of the following two
factors: 1) domestication at other places made it much easier for the gayal to affiliate
with yellow cattle or gaur, and the gayal lived semi-domestically, so it was possible to
give birth to lots of crossbreed descendants. 2) female crossbreed descendants were
able to reproduce, but male descendants could not [15]." Another paper cited by you
(Prabhu et al., 2019) also suggests a complex scenario when they state: "Our
phylogenetic analyses suggested the presence of two maternal lineages for mithun.
The occurrence of two types of mithun, one which is a descendant of gaur, while the
other which is a hybrid of mithun bull and cattle, has also been indicated by Dorji et
al. [11]. Similarly, Baig et al. [1] have reported three different haplotypes for mithun.
The recent studies [3, 5, 6, 26] based on whole genome sequencing and SNP

genotyping have supported all the three proposed hypotheses for the origin of mithun.
Therefore, the phylogenetic status of mithun remains contentious. Perhaps, the
evolutionary history of mithun is more complex as reported in the case of many other
livestock species. A comprehensive phylogenetic analysis with more number of
mithun, gaur and cattle samples particularly from India and China would improve the
current understanding of the evolutionary history of mithun."

Hence, the possibility of introgression from other closely related species (such as *Bos*
*gaurus*) is a genuine concern and is also of great importance to the Drung Cattle
conservation efforts.

**AU:** We sincerely thank you for this concern and helpful suggestions.

In the analysis of phylogeny and evolutionary history of Chinese gayal (Drung cattle),
we did consider the possibility of introgression of the *Bos* genus. We noticed that Wu
*et al.* has carried out a comprehensive genetic introgression analysis in the species of
the *Bos* genus using whole-genome sequencing, including taurine cattle, zebu, gayal,
gaur, banteng, yak, wisent and bison¹. Wu *et al.* performed a series of ABBA-BABA
tests and found evidence of genetic introgression signals among different members of
*Bos* genus. The frequency of shared identical-by-descent (IBD) was then calculated to
examine the genomic regions of potential introgression and detect the direction of
introgression. The result showed that genetic introgression was determined from
domestic cattle to wisent and yak, from zebu to gayal and banteng, and from yak to
Tibetan cattle in Figure 1 of the published paper¹. In their study, gayal samples were
collected from the same geographic area as our research. In view of the fact that the
relevant research results have been reported, we have cited the reference in the revised
manuscript. Please see lines 693-696.

As the reviewer's comments and many previous studies have inferred, the
phylogenetic position of gayal is still controversial. Our results indicated that *Bos*
*frontalis* was clearly distinct from *Bos taurus* and splits off from the genus *Bos*, which
is consistent with the statement that gayal is an independent species or subspecies
816 ^{1,22,23}.

Fig. 1: Phylogenetic tree and genetic introgression of species in the *Bos* genus (quoted from Wu DD
 *et al. Nat Ecol Evol*, 2018) ¹.

**References:**

- 1 Wu, D. D. *et al.* Pervasive introgression facilitated domestication and adaptation in the *Bos*
 species complex. *Nat Ecol Evol* **2**, 1139-1144, doi:10.1038/s41559-018-0562-y (2018).
 22 Ma, G. L. *et al.* Phylogenetic relationships and status quo of colonies for gayal based on
 analysis of Cytochrome b gene partial sequences. *Journal of Genetics and Genomics* **34**, 413-419,
 doi:10.1016/s1673-8527(07)60045-9 (2007).
 23 Baig, M. *et al.* Mitochondrial DNA diversity and origin of *Bos frontalis*. *Curr Sci India* **104**,
 115-120 (2013).

**Comment 3:**

See comments regarding main figures

**Figure 1** has a lot of data shown in a single figure. My suggestion is that this figure
 can be replaced by a more straightforward figure which is easier to follow. The
 transcriptome, methylome, etc., circles can be organized separately by chromosome in
 the supplements.

**AU:** Thanks for your suggestion. Figure 1 is to show our research methods and
 corresponding results from the perspectives of genome, transcriptome, methylation
 and other aspects, so that readers will get an overview of the article.

**Figure 3:** The gene order blocks shown on scaffold 60 in Figure 3C are not explicitly
identified in Figure 3A. Have you used the same color scheme to link figure 3A and
3C?

**AU:** The gene order blocks was not shown on Scaffold60 in Figure 3C, which is
presented in the **Supplementary Data 17**. The Figure 3C shows the satellite repeats
organization in centromere region. The same color between Figure 3C and Figure 3A
means different meanings.

**Figure 6:** The pathway shown in Figure 6A is based on the KEGG database, or is it
obtained from any particular source? Would you please provide a reference for the
connections and membrane localisations depicted in this figure?

The genes that are presumed missing in Figure 6B and Supplementary Table 19 could
be missing from the genome assembly. However, it is still possible that they are intact
in the genome. Have you searched the sequencing read datasets for these "missing
genes"? Do you find the pseudogene sequences? Would you please provide more
details?

**AU:** Yes, the pathway shown in Figure 6A was based on the KEGG database
including map04260, map04270, map04370 and map04810. We have provided more
detailed descriptions about Figure 6A, please see lines 579-582. As you are
concerned, we did search for the "missing gene" sequences from the sequencing read
datasets in the analysis of MYH gene family. A detailed description of each gene
member in the MYH gene family of *Bovinae* species is showed in the Supplementary
Data 22. The result showed that all 15 MYH gene members could be discovered in
diverse species of *Bovinae*, and no pseudogene sequence was found except that
*MYH16* only appeared in buffalo.

**Comment 4:**

"Genome-wide DNA methylation patterns". This section (lines 382-423) seems to
have very little connectivity to the text before and after this section. Can you please

improve this text? For instance, a tentative link between DNA methylation and
changes in the heart muscle is discussed later (in lines 616-620). Could you rewrite
this section to highlight this link? Alternatively, you could identify differentially
methylated regions between tissues and link them with changes in gene expression.
The role of DNA methylation in gene expression is widely acknowledged.

**AU:** Thank you for your recommendation. We have rewritten this section to highlight
connectivity between contexts. Transcriptome-based gene co-expression network
analysis identified gene expression specificity in the heart tissue of Drung cattle. To
further explore the epigenetic characteristics of tissue-specific, we profiled and
compared the genome-wide DNA methylation status in the heart and three other
tissues. Given that methylation analysis in gayal has not yet been reported, we showed
more details in the original draft in order to serve as a genetic resource for research on
other animals living in similar environments. Due to some less contextual findings,
we have removed these results in the revised manuscript to focus on the important
results of the lowest level in the heart with more non-CG DNA methylation than in
other tissues.

**Comment 5:**

You have mentioned, "Like zebu and yak, Drung cattle lost both MYH4 (clade IV)
and MYH6 (clade V) during gene family evolution. As a pseudogene, MYH16 gene
was missing in the other five selected species except buffalo. In addition, the clade
XVI with six gene members were only present in Drung cattle."

Changes in MYH gene content in Drung cattle is an important result that could
explain the difference in meat quality of gayal and other cattle. Hence, it needs to be
evaluated in detail. Have you searched the sequencing read datasets for these "missing
genes"? Do you find the pseudogene sequences? Are these pseudogene sequences
expressed in RNA-seq of any of the tissues? Would you please provide more details?

**AU:** Thanks for this concern. As you pointed out, we did search for the "missing
gene" sequences from the sequencing read datasets in the analysis of the MYH gene

family. A detailed description of each gene member in the MYH gene family of
 *Bovinae* species is showed in the Supplementary Data 20. The result showed that all
 15 MYH gene members could be discovered in diverse species of *Bovinae*, and no
 pseudogene sequence was found except that *MYH16* only appeared in buffalo. In
 addition, the expression of pseudogene sequences was not detected in the RNA-seq of
 any of the tissues, while other MYH genes had their respective expressions in tissues
 as shown in the following attached table.

Gene name	Gene ID	Heart	Lung	Liver	Spleen	Kidney	Rumen	Testis	Striploin	Rump	Tenderloin	Chuck	Neck	Abdominal fat	Subcutaneous fat
MYH1	CCG002410	17.0934	0	0	0	1.29752	0	0.1113	7011.32	1017.85	9513.28	7348.69	234.37	9.76699	8.77102
MYH2	CCG002411	15.1555	0	0	0	0.57831	0	0.01373	2194.24	5496.55	2914.99	2704.41	8183.8	11.4981	8.44718
MYH3	CCG002412	0.9201	1.30304	0.5508	0.78206	0.67508	1.24458	0.81871	1.43273	0.77462	1.72605	1.12156	1.87501	4.3205	3.27843
MYH7	CCG017036	0	0.37289	0.00654	0.05242	0.32998	0.03226	0.53608	1464.12	0	3222.01	3805.91	3337.19	16.2261	9.80155
MYH7B	CCG014602	807.68	0.68065	0.05279	1.53792	0.52719	0.90182	2.6634	99.8718	195.533	109.132	112.145	203.532	0.823734	0.888112
MYH8	CCG002408	1.26693	0	0	0.04945	0.011	0	0.72673	0.12394	0.11422	0.24592	1.26148	1.29091	0	0.0099181
MYH9	CCG004744	51.0662	190.862	98.1432	183.444	136.44	151.641	16.5525	79.4835	41.7751	44.8766	40.4587	26.3847	318.406	380.769
MYH10	CCG007004	8.32376	31.1071	12.8489	7.38551	21.4875	4.72548	6.33303	6.78322	7.10556	5.04992	4.87163	8.05197	23.5472	24.6466
MYH11	CCG005657	3.701	68.1742	17.5208	176.747	23.2851	284.911	31.3357	54.8381	21.9601	7.47925	14.4724	5.90753	21.2148	37.8919
MYH13	CCG002407	0	0	0	0	0	0	0.25312	0.0201	0.01907	0.13016	0.23494	0.18913	0	0
MYH14	CCG018391	90.1934	115.744	119.451	1.19521	49.9071	44.217	0.71537	86.0985	113.534	79.7212	127.151	115.866	87.5937	60.866
MYH15	CCG004151	0.01872	0	0	0.03312	0.01047	0.0103	0.74981	0.04185	0.89662	0.01964	0	1.33725	0.0093656	0

**Comment 6:**

Line 754-756: "PAML was used to identify positively selected genes from different
 species. The maximum-likelihood method was used to estimate dN, dS and dN/dS."
 More details regarding which tests were done using PAML are missing. Did you
 perform branch tests? Supplementary Data 6 provides P-values. Are these p-values the
 results of Likelihood ratio tests? Are you able to give the likelihood values of the tests
 done?

**AU:** Thanks for your suggestion. The detailed method was described in the Methods.
 "PAML (v4.9) was employed to detect genes under positive selection for each single
 copy orthologous gene family. For branch-site model analysis, we used all single copy
 orthologous gene regions of eleven mammals and selected Drung cattle in the
 foreground position on the phylogeny. We then compared the result of null model
 ($\omega=1$, NSsites=2, $\omega=1$ and fix_omega=1) with the alternative model
 ($\omega=1.5$, NSsites=2, $\omega=1.5$ and fix_omega=0) using likelihood rate statistics
 method of CODEML software for each branch-site model of every orthologous gene.
 χ^2 distribution was used to evaluate the significance ($p < 0.05$) of the likelihood ratio
 statistic." Please see lines 1006-1019.

**Minor comments:**

1) line 3: "...specie that mainly inhabits the hill-forests of "Grand Canyon of the East",
Yunnan..."

**AU: Revised.**

2) line 51: "...has grown from 77 individuals at risk of extinction in 1,986 to nearly
3,000 heads now..."

**AU: Revised.**

3) line 58: "In term of biological classification, Drung cattle belongs to species *Bos*
*frontalis*..."

**AU: Revised.**

4)line 70: "..altitude of 1,170-4,964 meters, which gives them unique physiological
features that.."

**AU: Revised.**

5) line 71-73: "Studies have showed that the level of hemoglobin of Drung cattle, as
well as the number of red blood cells and white blood cells, were equivalent to those
of yaks living on the Tibetan Plateau.". No citations are provided here. Which studies
are you referring to in this sentence?

**AU: Revised. Please see line 77.**

6) line 149-156: "In comparison with taurine cattle genome, Drung cattle genome
overrepresented ~40% LINE-L1 repeats, and had ~87% more LINE-RTE(BovB)
repeats (702,019 in *Bos frontalis* and 376,067 in *Bos taurus*)#;instead, both LINE-L2
and LINE-CR1 decreased by approximately 1.5-fold, and SINE-BovA repeats had
a >88% reduction (976,749 in *Bos frontalis* and 1,839,497 in *Bos taurus*) than that in
taurine cattle#, indicating that as the ruminant- or cattle-specific repeats, SINE-BovA

expanded primarily in the *Bos taurus* genome, whereas LINE-RTE(BovB) expanded
specifically in the Drung cattle."

Consider splitting and rewriting the above into shorter sentences at the hash (#)
annotations in the line above.

**AU:** Revised.

7) You note differences in repeat content of the genome assemblies and attribute this
to biological differences. However, repeat regions are known to be difficult to
assemble regions of the genome. Could these differences be a result of differences in
how well these repeats are assembled in these genomes? Have you done any due
diligence to rule out the possibility of assembly quality differences confounding the
biology? For example, can you demonstrate few examples of species-specific repeats
are supported by raw read datasets?

**AU:** First, compared with the previously published reference genome assembly level
of Chinese gayal, the various indicators of the Drung cattle are higher; secondly, it is
at a better compared with the bovine genome assembly level.

8) line 201: "...gene families were contracted in gayal (Fig. 2a). Among the four
species of genus *Bos*, gayal showed more events of gene family expansion with a 50-
100 increase in..".

In the above sentence, do you mean genus *Bos*?

**AU:** The four species of genus *Bos* contain *Bos taurus*, *Bos indicus*, *Bos grunniens*
and *Bubalus bubalis*.

9) Line 232-234: "Among our gayal 20,181 annotated genes, we found 885 genes
belonged to the OR supergene family, of which 22 OR genes originating from SD
were exclusive to Drung cattle."

You refer to *Bos frontalis* as both gayal and Drung cattle in the above sentence. Would
you please use consistent terms throughout the manuscript? Either use just Drung

cattle or gayal. Alternatively, you can write it as gayal/Drung cattle throughout the
manuscript.

**AU:** Revised, we write it as Drung cattle throughout the manuscript.

10) Line 578-580: "The strong resistance of cold and humidity as well as high stress
tolerance to the environment are the remarkable features of adaptability in Drung
cattle."

Change the above sentence to "The strong resistance to cold and humidity as well as
high-stress tolerance to the environment are the remarkable features of adaptability in
Drung cattle."

**AU:** Revised.

11) Line 695-696: "Clean reads were mapped to the Bos frontalis genome and
Btau_5.0.1 assembly using Bowtie and TopHat, respectively." Why are two different
read mappers used for the two genomes? Have you used bowtie read mapper for the
transcriptome of Bos frontalis? Would you please correct this sentence?

**AU:** In this study, the Bowtie software was used to build-index for genomes instead
of reads mapping. The TopHat software was used to aligns RNA-Seq reads to
genomes.

12) Line 699-700: "Repeat sequences were identified with de novo-based and
homolog-based methods. The de novo gayal repeat library were build using Piler139
and RepeatModeler programmers."

Change the above sentence to "Repeat sequences were identified with de novo-based
and homolog-based methods. The de novo gayal repeat library was built using Piler
and RepeatModeler programs."

**AU:** Revised.

13) Line 748: "Calibration time was obtained from the TimeTree database.". How was

the calibration time used to estimate split times in the phylogenetic trees constructed
by PhyML? Are the split times from the time tree phylogeny?
**AU:** Thanks for your suggestion. We have described the construction of the
phylogenetic tree and the calculation of the divergence time in the Supplementary
Note. “To study the evolutionary history of *Bos frontalis*, we selected six species in
the Bovinae subfamily (*Bos frontalis*, *Bos taurus*, *Bos indicus*, *Bison bison*, *Bos*
*mutus*, and *Bubalus bubalis*,) and other five mammals (*Ovis aries*, *Capra hircus*,
*Pantholops hodgsonii*, *Ceratotherium simum simum*, and *Homo sapiens*) as the
outgroup. The protein-coding genes of the ten mammalian species that downloaded
from NCBI and our *Bos frontalis* assembly genome were used to define gene families.
TreeFam (v9) clustered all the genes grouping into orthology/ paralogy and provided
the evolutionary history of genes. Only the longest ORF was chosen to represent each
gene, and ORF of genes encoding <30 amino acids were filtered out. Among the
eleven species, the number of genes in families ranged from 8,643 (*B. frontalis*) to
9,648 (*H. sapiens*) (Supplementary Table 4). The 2,301 single-copy gene families
obtained from gene family clusters analysis were used to construct phylogenetic trees
based on the maximum likelihood method by the PhyML (v3.0) software. The
MCMCtree program implemented in the Phylogenetic Analysis by Maximum
Likelihood (PAML) package was employed to estimate the divergence time.
Calibration time was obtained from the TimeTree database. MCMCtree was run to
sample 200,000 times, with sample frequency set to 2, after a burn-in of 20,000
iterations. The divergence time of each node was calibrated against the fossil
calibration times for the *Homo sapiens-Bos taurus* divergence (95.3–113 Mya) and
*Bos taurus-Ovis aries* (18.3–28.5 Mya)”. In addition, the divergence time was the
time tree phylogeny. Estimates of divergence time and its interval based on sequence
identity are indicated at each node.

14) Few other potential typos that have been corrected:

**AU:** Revised.

Line 7: environmental adaption have not been

**AU:** Revised.

Line 47: Drung cattle were included

**AU:** Revised.

Line 50: the Chinese

**AU:** Revised.

Line 52: that Drung cattle were domesticated

**AU:** Revised.

Line 68: *Bos frontalis* had relatively lower completeness

**AU:** Revised. Please see lines 100.

Line 69: Drung cattle inhabit the typical alpine valleys

**AU:** Revised.

Line 79: the gayal remains hitherto unclear

**AU:** Revised. Please see lines 89-90.

Line 135: autosomes

**AU:** Revised. Please see lines 177.

Line 303: we propose an evolutionarily conserved

**AU:** Revised. Please see lines 414.

Line 315: carries out

**AU:** Revised.

Line 328: had a strong association

**AU:** Revised. Please see lines 439.

Line 329: with certain tissues

**AU:** Revised. Please see lines 442.

Line 330: functional modules had corresponding unique tissues, respectively

**AU:** Revised. Please see lines 443.

Line 333: modules are fundamental

**AU:** Revised. Please see lines 446.

Line 334: in the heart relative to other

**AU:** Revised. Please see lines 448.

Line 371: hub genes were constructed

**AU:** Revised. Please see lines 518.

Line 373: involved in the circulatory system and muscle

**AU:** Revised. Please see lines 519.

Line 405: in the heart

**AU:** Revised. Please see lines 533.

Line 409: preferences for methylation were observed

**AU:** Revised.

Line 414: tends to use adenine (A) in non-CG contexts preferentially
**AU:** Revised.
Line 423: lowest in the lung and
**AU:** Revised. Please see lines 566.
Line 438: voltage-gated
**AU:** Revised. Please see lines 589.
Line 450: Drung cattle were positively selected
**AU:** Revised. Please see lines 602.
Line 455: the combined effects
**AU:** Revised. Please see lines 610.
Line 492: To date, the origin
**AU:** Revised. Please see lines 657.
Line 496: from the crossing of wild gaur
**AU:** Revised. Please see lines 661.
Line 497: . The research on [full stop missing before the beginning of a new sentence]
**AU:** Revised. Please see lines 663.
Line 518: at least one million years earlier
**AU:** Revised. Please see lines 691.
Line 549: So far, there is
**AU:** Revised. Please see lines 728.

Line 556: which frequently occurred during

**AU:** Revised. Please see lines 737.

Line 568: promotes the evolution of the Bovidae

**AU:** Revised. Please see lines 752.

Line 570: understanding of the karyotype reorganization

**AU:** Revised. Please see lines 755.

Line 579: high-stress tolerance to the environment, are the

**AU:** Revised.

Line 586: olfactory receptors (OR)

**AU:** Revised. Please see lines 773.

Line 600: odor perception in food-seeking

**AU:** Revised.

Line 603: crucial to understand the evolutionary responses of organisms better

**AU:** Revised.

Line 604: Frisch et al.

**AU:** Revised. Please see lines 796.

Line 607: cellular responses to higher-order phenotypes

**AU:** Revised.

Line 616: expressed genes of Drung cattle

**AU:** Revised.
Line 623: torrid, humid, and rainy climates.
**AU:** Revised.
Line 635: histology gives us
**AU:** Revised.
Line 636: biological processes related to
**AU:** Revised.
Line 641: the expansion of the MYH gene family
**AU:** Revised. Please see lines 841.
Line 645: significant differences between affected and unaffected animals
**AU:** Revised. Please see lines 846.
Line 656: insights into the molecular mechanism
**AU:** Revised. Please see lines 861.
Supplementary Material
Line 190: TRF version not mentioned
**AU:** Revised. Please see lines 197.
Line 209: Splice junction
**AU:** Revised. Please see lines 218.
Line 233: orthology/paralogy
**AU:** Revised. Please see lines 241.

Line 385: magenta module

**AU:** Revised. Please see lines 388.

**References**

- Wu, D. D. *et al.* Pervasive introgression facilitated domestication and adaptation in the
Bos species complex. *Nat Ecol Evol* **2**, 1139-1144, doi:10.1038/s41559-018-0562-y
(2018).
- Schueler, M. G., Higgins, A. W., Rudd, M. K., Gustashaw, K. & Willard, H. F. Genomic and
genetic definition of a functional human centromere. *Science* **294**, 109-115, doi:DOI
10.1126/science.1065042 (2001).
- Sullivan, L. L. & Sullivan, B. A. Genomic and functional variation of human centromeres.
*Exp Cell Res* **389**, 111896, doi:10.1016/j.yexcr.2020.111896 (2020).
- Escudeiro, A., Adegas, F., Robinson, T. J., Heslop-Harrison, J. S. & Chaves, R.
Conservation, Divergence, and Functions of Centromeric Satellite DNA Families in the
Bovidae. *Genome Biology and Evolution* **11**, 1152-1165, doi:10.1093/gbe/evz061 (2019).
- Suntronpong, A. *et al.* CENP-B box, a nucleotide motif involved in centromere
formation, occurs in a New World monkey. *Biol Letters* **12**, doi:ARTN 20150817
10.1098/rsbl.2015.0817 (2016).
- Balzano, E. & Giunta, S. Centromeres under Pressure: Evolutionary Innovation in Conflict
with Conserved Function. *Genes-Basel* **11** (2020).
- Thakur, J., Packiaraj, J. & Henikoff, S. Sequence, Chromatin and Evolution of Satellite
DNA. *Int J Mol Sci* **22**, doi:ARTN 4309
10.3390/ijms22094309 (2021).
- Suzuki, Y., Myers, E. W. & Morishita, S. Rapid and ongoing evolution of repetitive
sequence structures in human centromeres. *Sci Adv* **6** (2020).
- Morozov, V. M., Giovinnazzi, S. & Ishov, A. M. CENP-B protects centromere chromatin
integrity by facilitating histone deposition via the H3.3-specific chaperone Daxx.
*Epigenet Chromatin* **10** (2017).
- Gamba, R. & Fachinetti, D. From evolution to function: Two sides of the same CENP-B
coin? *Experimental Cell Research* **390** (2020).
- Barra, V. & Fachinetti, D. The dark side of centromeres: types, causes and consequences
of structural abnormalities implicating centromeric DNA. *Nat Commun* **9** (2018).
- Rutland, C. S. *et al.* Knockdown of embryonic myosin heavy chain reveals an essential
role in the morphology and function of the developing heart. *Development (Cambridge,
England)* **138**, 3955-3966, doi:10.1242/dev.059063 (2011).
- Singer, E. S. *et al.* Characterization of clinically relevant copy-number variants from
exomes of patients with inherited heart disease and unexplained sudden cardiac death.
*Genet Med* **23**, 86-93 (2021).
- 14 Ma, Y. S. *et al.* Proteogenomic characterization and comprehensive integrative genomic
analysis of human colorectal cancer liver metastasis (vol 17, 139, 2018). *Mol Cancer* **18**
(2019).
- Molck, M. C. *et al.* Genomic imbalances in syndromic congenital heart disease. *J Pediat-
Brazil* **93**, 497-507 (2017).
- Haraksingh, R. R. *et al.* Exome sequencing and genome-wide copy number variant
mapping reveal novel associations with sensorineural hereditary hearing loss. *BMC
genomics* **15** (2014).
- Zhang, L. Z. *et al.* Detection of copy number variations and their effects in Chinese bulls.

*BMC genomics* **15** (2014).
Xu, Y. *et al.* Associations of MYH3 gene copy number variations with transcriptional
expression and growth traits in Chinese cattle. *Gene* **535**, 106-111 (2014).
Hiendleder, S., Lewalski, H. & Janke, A. Complete mitochondrial genomes of *Bos taurus*
and *Bos indicus* provide new insights into intra-species variation, taxonomy and
domestication. *Cytogenet Genome Res* **120**, 150-156, doi:10.1159/000118756 (2008).
Pramod, R. K. *et al.* Complete mitogenome reveals genetic divergence and phylogenetic
relationships among Indian cattle (*Bos indicus*) breeds. *Anim Biotechnol* **30**, 219-232,
doi:10.1080/10495398.2018.1476376 (2019).
Wang, M. S. *et al.* Draft genome of the gayal, *Bos frontalis*. *GigaScience*,
doi:10.1093/gigascience/gix094 (2017).
22 Ma, G. L. *et al.* Phylogenetic relationships and status quo of colonies for gayal based on
analysis of Cytochrome b gene partial sequences. *Journal of Genetics and Genomics* **34**,
413-419, doi:10.1016/s1673-8527(07)60045-9 (2007).
23 Baig, M. *et al.* Mitochondrial DNA diversity and origin of *Bos frontalis*. *Curr Sci India* **104**,
115-120 (2013).

REVIEWERS' COMMENTS:

Reviewer #1 (Remarks to the Author):

Although authors revised most parts of the manuscript and also tried to address my previous concerns, I feel that the present manuscript needs to be revised further in the following parts:

- 1) The logical connections between DNA methylation and other parts are still confusing without a clear aim and reason.
- 2) Many parts of the discussion part are repeating those results without clear logical reasons.
- 3) English has not been improved, especially introduction and discussion parts.
- 4) Too many references are not cited correctly, some repeats and some lost and some unrelated to contents.

Minor issues:

1. Line 62-63: Please check the syntax
2. The resolution of Fig. 1, Fig. 2 and Fig. 4 is too low to see the contents of the figure, please provide a higher resolution image.
3. Line 227, 231: '1100 rRNAs, 1948 snRNAs'; '2,301 single-copy gene families' uniform number writing format.
4. Line 241: (Bos, Bison and Bubalus), which taxon does Bos refer to.
5. Line 382-383: Please change '2 blocks...8 blocks on BTA28 and 3 blocks on', 'two blocks..., eight blocks on BTA28 and three blocks on'. '3 hub genes and 1 Drung-specific gene '

Reviewer #2 (Remarks to the Author):

The manuscript has been drastically improved. In particular, the introduction section has been carefully revised by removing the unnecessary descriptive statements about Drung cattle and explaining the problem of the existing Drung cattle assembly. Missing details about the methods (parameters, software versions, etc) have been provided in the text and supplementary notes. The limitation of the study, using only one biological replicate, has been added to the discussion. Globally, considerable efforts have been made to address the issues raised in the original review, resulting in a different and much better manuscript. Congratulations to the authors for this work.

The only remaining modifications I would suggest are (see below for details):

- Authors mentioned that detailed materials and methods are put in the supplementary notes. I would suggest putting "please see supplementary notes" in parenthesis in front of the relevant sentences in methods.

Reviewer #3 (Remarks to the Author):

The authors have answered all the comments.

Many more details about the methodology have also been provided.

REVIEWERS' COMMENTS:

Reviewer #1 (Remarks to the Author):

Although authors revised most parts of the manuscript and also tried to address my previous concerns, I feel that the present manuscript needs to be revised further in the following parts:

1) The logical connections between DNA methylation and other parts are still confusing without a clear aim and reason.

AU: Thank you for pointing out this. In view of the reviewer's concerns, we have removed some of the less logical results in the methylation study with other sections, and retained a brief description of methylation integrated into the genome profile.

Please see lines 164-169.

2) Many parts of the discussion part are repeating those results without clear logical reasons.

AU: We have revised the repeated description of the conclusion in the discussion and improved the logic of the context.

3) English has not been improved, especially introduction and discussion parts.

AU: Thanks for your suggestion. We have further improved the English language of the manuscript, especially the introduction and discussion parts. The revised manuscript has been proofread by an English language editing agency (American Journal Experts, AJE).

4) Too many references are not cited correctly, some repeats and some lost and some unrelated to contents.

AU: Revised.

Minor issues:

1. Line 62-63: Please check the syntax

AU: Revised. Please see lines 76.

2.The resolution of Fig. 1, Fig. 2 and Fig. 4 is too low to see the contents of the figure, please provide a higher resolution image.

AU: Revised.

3.Line 227, 231: ‘1100 rRNAs, 1948 snRNAs’; ‘2,301 single-copy gene families’ uniform number writing format.

AU: Revised. Please see line 203.

4.Line 241: (Bos, Bison and Bubalus), which taxon does Bos refer to.

AU: *Bos* refer to the genera *Bos*. Bovids, belong to the *Bovini* tribe, which is composed of the genera *Bos*, *Bison*, and *Bubalus*.

5.Line 382-383: Please change ‘2 blocks...8 blocks on BTA28 and 3 blocks on’ to ‘two blocks..., eight blocks on BTA28 and three blocks on’. ‘3 hub genes and 1 Drug-specific gene’

AU: Revised. Please see lines 335-336, line 453.

Reviewer #2 (Remarks to the Author):

The manuscript has been drastically improved. In particular, the introduction section has been carefully revised by removing the unnecessary descriptive statements about Drug cattle and explaining the problem of the existing Drug cattle assembly. Missing details about the methods (parameters, software versions, etc) have been provided in the text and supplementary notes. The limitation of the study, using only one biological replicate, has been added to the discussion. Globally, considerable

efforts have been made to address the issues raised in the original review, resulting in a different and much better manuscript. Congratulations to the authors for this work.

The only remaining modifications I would suggest are (see below for details):

- Authors mentioned that detailed materials and methods are put in the supplementary notes. I would suggest putting “please see supplementary notes” in parenthesis in front of the relevant sentences in methods.

AU: We have integrated the methods described in the Supplementary Notes into the main text.